# Carboxysome encapsulation of the $CO_2$-fixing enzyme Rubisco in tobacco chloroplasts

Benedict M. Long [1], Wei Yih Hee [1], Robert E. Sharwood[2], Benjamin D. Rae [2], Sarah Kaines [1], Yi-Leen Lim [1], Nghiem D. Nguyen [2], Baxter Massey [1], Soumi Bala[2], Susanne von Caemmerer [1,2], Murray R. Badger[1,2] & G. Dean Price[1,2]

A long-term strategy to enhance global crop photosynthesis and yield involves the introduction of cyanobacterial $CO_2$-concentrating mechanisms (CCMs) into plant chloroplasts. Cyanobacterial CCMs enable relatively rapid $CO_2$ fixation by elevating intracellular inorganic carbon as bicarbonate, then concentrating it as $CO_2$ around the enzyme Rubisco in specialized protein micro-compartments called carboxysomes. To date, chloroplastic expression of carboxysomes has been elusive, requiring coordinated expression of almost a dozen proteins. Here we successfully produce simplified carboxysomes, isometric with those of the source organism *Cyanobium*, within tobacco chloroplasts. We replace the endogenous Rubisco large subunit gene with cyanobacterial Form-1A Rubisco large and small subunit genes, along with genes for two key α-carboxysome structural proteins. This minimal gene set produces carboxysomes, which encapsulate the introduced Rubisco and enable autotrophic growth at elevated $CO_2$. This result demonstrates the formation of α-carboxysomes from a reduced gene set, informing the step-wise construction of fully functional α-carboxysomes in chloroplasts.

[1] Realizing Increased Photosynthetic Efficiency (RIPE), The Australian National University, 134 Linnaeus Way, Acton, ACT 2601, Australia. [2] Australian Research Council Centre of Excellence for Translational Photosynthesis, Research School of Biology, The Australian National University, 134 Linnaeus Way, Acton, ACT 2601, Australia. These authors contributed equally: Benedict M. Long, Wei Yih Hee. Correspondence and requests for materials should be addressed to B.M.L. (email: ben.long@anu.edu.au)

Photosynthetic $CO_2$ fixation via ribulose-1,5-bisphosphate carboxylase/oxygenase (Rubisco) is the primary input of carbon into crop biomass. However, Rubisco-mediated $CO_2$ fixation in $C_3$ chloroplasts is catalytically slow, competitively inhibited by oxygen[1] and, from an agricultural stand-point, makes inefficient use of water and combined nitrogen[2]. These latter inefficiencies are driven by passive acquisition of $CO_2$ from the air (leading to water loss via open stomata) and by large investment in Rubisco (up to 50% of leaf protein) to overcome its poor kinetics[3]. A suggested approach to increase $CO_2$ fixation, minimize water-loss and decrease investment in Rubisco is to translate essential components of the cyanobacterial $CO_2$-concentrating mechanism (CCM) into $C_3$ crops[4–6].

The cyanobacterial CCM is a single-cell, bipartite system that first generates a high intracellular bicarbonate ($HCO_3^-$) pool through action of membrane-bound inorganic carbon ($C_i$) transporters and $CO_2$-converting complexes[7–9] (Fig. 1a). This $HCO_3^-$ pool is then utilized by subcellular micro-compartments called carboxysomes, which encapsulate the cell's complement of

Rubisco[10]. The carboxysome's outer protein shell enables diffusional influx of $HCO_3^-$ and RuBP, where the former is converted to $CO_2$ by a localized carbonic anhydrase (CA). Physiological evidence[11] and mathematical models[12,13] suggest carboxysomes resist $CO_2$ efflux, resulting in concentration of $CO_2$ around Rubisco. Cyanobacterial carboxysomes possess high-catalytic-turnover, but low-$CO_2$-specificity Rubisco enzymes[1]. When the intracellular $HCO_3^-$ pool is elevated, a high $CO_2$ environment can be generated inside the carboxysome, overcoming this low specificity and enabling rapid $CO_2$ fixation with reduced inhibition by oxygen[14].

An engineering strategy to generate a chloroplastic CCM in crop plants (Fig. 1b) relies on transfer of genes encoding $HCO_3^-$ transporters, directed to the chloroplast inner-envelope membrane (IEM), to generate an elevated stromal $HCO_3^-$ pool, and genes encoding the carboxysome and its Rubisco[4,6]. Active $HCO_3^-$ transporters in the chloroplast IEM alone are expected to improve photosynthesis due to the elevation of $CO_2$ concentrations around Rubisco[15,16]. Notably, either a carboxysome-

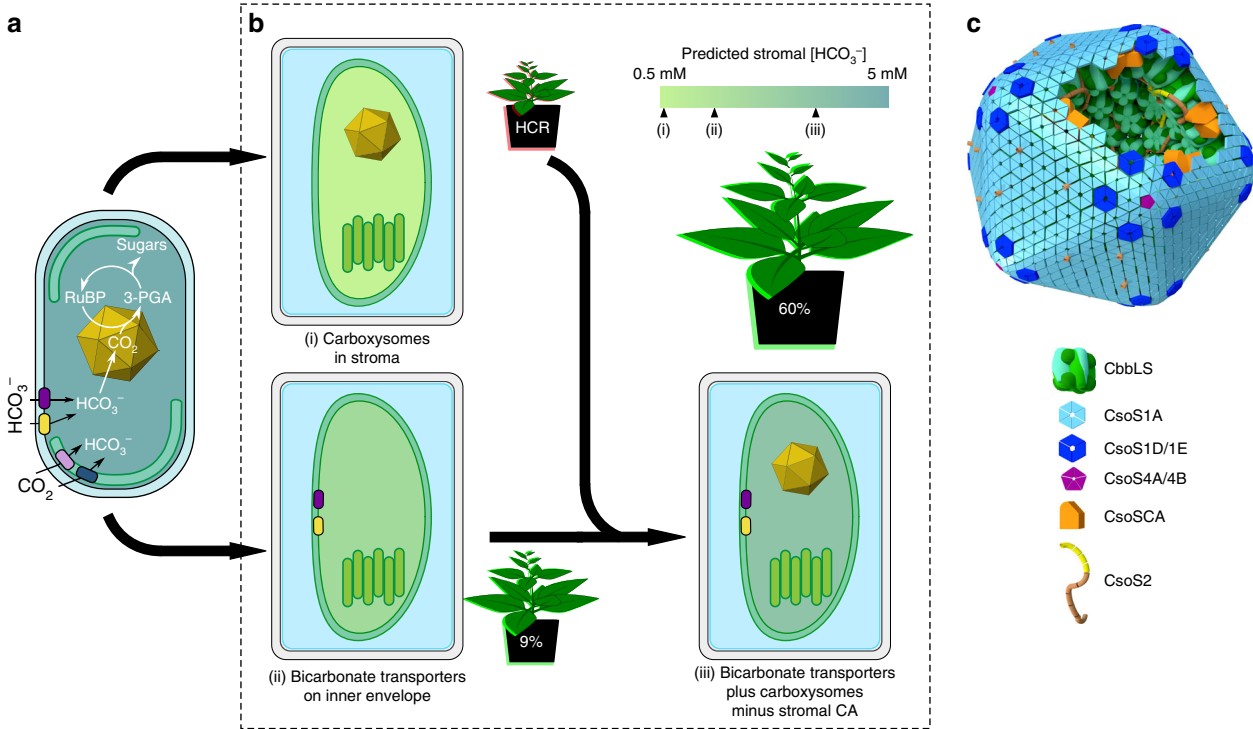

**Fig. 1** Cyanobacterial CCM components for improved photosynthesis. Cyanobacterial CCMs (**a**) use bicarbonate ($HCO_3^-$) and $CO_2$ pumps on the plasma and thylakoid membranes, respectively, to elevate cytosolic $HCO_3^-$. This $HCO_3^-$ pool is utilized by icosahedral-shaped Rubisco microcompartments called carboxysomes (yellow icosahedron), where $HCO_3^-$ is converted to $CO_2$ by localized carbonic anhydrase (CA) and accumulates due to resistive $CO_2$ efflux. The carboxysome encapsulates the cell's complement of a high-catalytic-rate Rubisco, which operates at close to $V_{max}$, converting ribulose-1,5-bisphosphate (RuBP) to 3-phosphoglycerate (3-PGA) within the Calvin cycle. This mechanism leads to efficient photosynthesis and reduced nitrogen allocation compared to $C_3$ plants. A strategy to improve $C_3$ photosynthesis in plant cells (**b**, represented here as rectangular structures containing dual-membrane chloroplasts) using the cyanobacterial CCM recommends independent transfer of carboxysomes containing their cognate Rubisco to the chloroplast stroma (i) and $HCO_3^-$ pumps (ii) to the chloroplast inner-envelope membrane. Successful transfer of $HCO_3^-$ transporters alone (ii) should generate a moderately elevated stromal $HCO_3^-$ pool (indicated by the change in colour shading of the chloroplast) and has a predicted photosynthetic improvement of 9%[15] due to the resulting net elevation of $CO_2$ near Rubisco. Expression of functional carboxysomes (i), or just their cognate Rubisco, in the chloroplast stroma should lead to a high $CO_2$-requiring (HCR) phenotype due to the characteristically high $K_M$ and low specificity for $CO_2$ of carboxysomal Rubiscos and absence of an elevated $HCO_3^-$ or $CO_2$ pool. However, in combination (iii), generation of high stromal $HCO_3^-$ pool in the presence of functional carboxysomes, with stromal CA eliminated, is predicted to generate a stromal $HCO_3^-$ concentration approaching 5 mM[16] and to increases in $CO_2$ fixation and yield of up to 60%[15]. **c** Carboxysomes of *Cyanobium* PCC7001, used in this study, consist of many thousands of polypeptides, arranged in an icosahedral structure. In this model, a single layer of shell-bound Rubisco (CbbLS, green) is shown, with carboxysomal CA (orange). CsoS2 (yellow/brown) interlinks Rubisco and the shell made predominantly of CsoS1A hexamers (light blue). These and ancillary shell proteins (CsoS1D and CsoS1E, dark blue) enable substrate transport via central pores. Pentameric vertex proteins (CsoS4AB, purple) complete the structure

encapsulated or free cyanobacterial Rubisco in $C_3$ plant chloroplasts will effectively lead to high $CO_2$ requirement for growth because cyanobacterial Rubiscos have low affinity and specificity for $CO_2$[1]. Stromal $HCO_3^-$ pools in $C_3$ plants grown in air approximate 0.5 mM[17], but the cyanobacterial cytoplasm reaches concentrations between 5 and 20 mM[18], despite low external $C_i$[19], to drive the CCM. In combination, a high stromal $HCO_3^-$ pool generated by active $HCO_3^-$ transporters and a fully functional carboxysome where $CO_2$ can be elevated could improve $C_3$ plant $CO_2$ fixation and yield up to 60%[15,20]. This improvement would provide savings in energy costs for the plant and both nitrogen and carbon investment in the $CO_2$ fixation machinery[5]. Elimination of the native stromal CA and $C_3$ Rubisco to further improve the accumulation of $HCO_3^-$ within the stroma is required to realize an optimal functioning chloroplastic CCM[4,6,16].

Within this proposed engineering strategy, construction of the carboxysome is particularly challenging due to genetic and protein-organizational complexity and requirements for functionality; some carboxysomes require coordinated expression of 13 genes[4]. Carboxysomes are a subset of proteinaceous bacterial microcompartments (BMCs[21]), with specialized $CO_2$ anabolic function[22]. Two carboxysome types have arisen through convergent evolution: α-carboxysomes encapsulate Form-1A Rubisco in proteobacteria and some unicellular cyanobacteria, and β-carboxysomes encapsulate the plant-like Form-1B Rubisco in the remaining cyanobacteria[10,23]. Noting that the biogenesis and composition of each carboxysome type is unique[10], components of the β-type lumen have been successfully expressed in *Nicotiana tabacum* (hereafter tobacco) chloroplasts[24]. This showed that cyanobacterial Form-1B Rubisco could be successfully expressed and cross-linked with CcmM35[25] to form large aggregates in the chloroplast[24]. Additionally, transient expression studies showed that carboxysome shell proteins could interact and form structures within chloroplasts[26]. However, these attempts could not generate structural carboxysomes nor encapsulate Rubisco, key requirements to generate $CO_2$ around Rubisco and for overall CCM functionality[5]. While carboxysomes have been heterologously expressed in bacterial systems[27,28], there are currently no reports of α- or β-carboxysome biogenesis in eukaryotic systems.

In this study, we designed simplified α-carboxysomes inspired by those from *Cyanobium marinum* PCC7001 (hereafter *Cyanobium*). *Cyanobium* carboxysomes likely consist of a protein shell primarily made up of CsoS1A, interspersed with proteins CsoS1D and/or CsoS1E[29] (Fig. 1c). Together, these proteins are envisaged to provide a selectively permeable shell, allowing $HCO_3^-$ and RuBP into the carboxysome and 3-PGA release[30] but limiting $CO_2$ efflux[13,31]. Within the carboxysome, CsoSCA, a CA on the inner shell surface[32], converts accumulated $HCO_3^-$ to $CO_2$. Rubisco (comprising CbbL and CbbS subunits) is likely anchored to the shell via CsoS2[33], which arises as two isoforms from one gene in many α-carboxysomal species but only one isoform in *Cyanobium*[29,33]. The pentameric vertex proteins (CsoS4A and CsoS4B) complete the icosahedral structure, since their absence can lead to elongated carboxysomes with aberrant function[34].

While a complete *Cyanobium* carboxysome requires at least nine polypeptides, generating plants containing genes for all these proteins would increase the risk of unforeseen errors in expression, transgene stability and carboxysome biogenesis. Instead, a bottom–up approach to carboxysome construction in a eukaryotic host seems a more practical proposal. Given the self-organizing nature of CsoS1A[35,36], and the carboxysome-organizing role of CsoS2 in complex with both CsoS1A and Rubisco[33], we hypothesized that these components alone may provide a minimal set of proteins for a simplified carboxysome

assembly design, with the potential for Rubisco encapsulation upon their co-expression. We constructed multigene cassettes for tobacco chloroplast transformation that contained genes for *Cyanobium* Rubisco large subunit (LSU, *cbbL*) and small subunit (SSU, *cbbS*) or these genes in combination with those for α-carboxysome proteins CsoS1A and CosS2 (Fig. 2). These genetic expression constructs were introduced into the tobacco plastome where they replaced the endogenous Rubisco LSU gene.

Here we report an example of structural carboxysomes, encapsulating a cyanobacterial Form-1A Rubisco, expressed in plant chloroplasts. The primary outcome is the formation of structurally identifiable and purifiable carboxysome structures, formed with Rubisco and just two shell proteins (CsoS1A and CsoS2). This provides a proof-of-concept for the construction of complete and functional carboxysomes within the chloroplast.

## Results

**Generation of transgenic plants**. Chloroplasts of the *Rhodospirillum rubrum*-tobacco (cm^trL) master line[37] carrying the single-subunit Rubisco (RbcM) were transformed with gene expression cassettes for *Cyanobium* Rubisco only (CyLS) or CyLS in concert with the carboxysome proteins CsoS1A and CsoS2 (CyLS-$S_1S_2$; Fig. 2) using biolistic bombardment. Transgene expression cassettes also carried the spectinomycin selection marker gene, *aadA*, and transformants were recovered through selective tissue culture. Successful plantlets were subsequently grown in soil at 2% (*v/v*) $CO_2$ to flowering and seed collection. Southern blot analysis using a DNA probe specific to common sequence in wild-type, cm^trL and transgenic plants confirmed the presence of DNA fragments of the anticipated size (Fig. 2). In addition, the complete loss of the *R. rubrum rbcM* DNA fragment indicated that transformant lines were homoplasmic (Fig. 2).

**Protein content in transformed tobacco leaves**. To determine whether successful introduction of transgenes led to production of cyanobacterial proteins, we conducted sodium dodecyl sulfate-polyacrylamide gel electrophoresis (SDS-PAGE) and immunoblot analysis of protein in leaf extracts. This confirmed the presence of the LSU and SSU of *Cyanobium* Rubisco in CyLS plant leaves, along with both carboxysome structural proteins CsoS1A and CsoS2 in CyLS-$S_1S_2$ leaves (Fig. 3a). Using antibodies specific to the *R. rubrum* Rubisco (RbcM) in the the cm^trL host plants, we confirmed the absence of RbcM from CyLS and CyLS-$S_1S_2$ leaves. An antibody which detects both the tobacco RbcL and *Cyanobium* CbbL proteins confirmed that CyLS and CyLS-$S_1S_2$ lines contained only the *Cyanobium* Rubisco. We also noted the presence of small quantities of the nuclear-encoded tobacco SSU (RbcS) in CyLS-$S_1S_2$ leaf material and the successful expression of *Cyanobium* CbbS in transformants.

**Leaf ultrastructure reveals carboxysome formation**. To determine whether carboxysome protein expression led to ultrastructural chloroplast changes, we carried out transmission electron microscopy (TEM) of ultrathin leaf sections. Chloroplasts of CyLS plants showed no observable abnormalities, with chloroplasts of these plants typical of those found in wild-type leaves (Fig. 3b). CyLS-$S_1S_2$ chloroplasts, however, contained multiple electron-dense particles (Fig. 3c, d), similar to *Cyanobium* α-carboxysomes[18]. Closely packed, localized clusters of geometric structures, approximately 100 nm in diameter, were observed in most chloroplast sections of CyLS-$S_1S_2$ plants.

Presuming that the structures observed in CyLS-$S_1S_2$ chloroplasts were carboxysomes, a variation of an α-carboxysome purification technique[38] was used to isolate particles from leaf tissue. The same method was also used to isolate carboxysomes

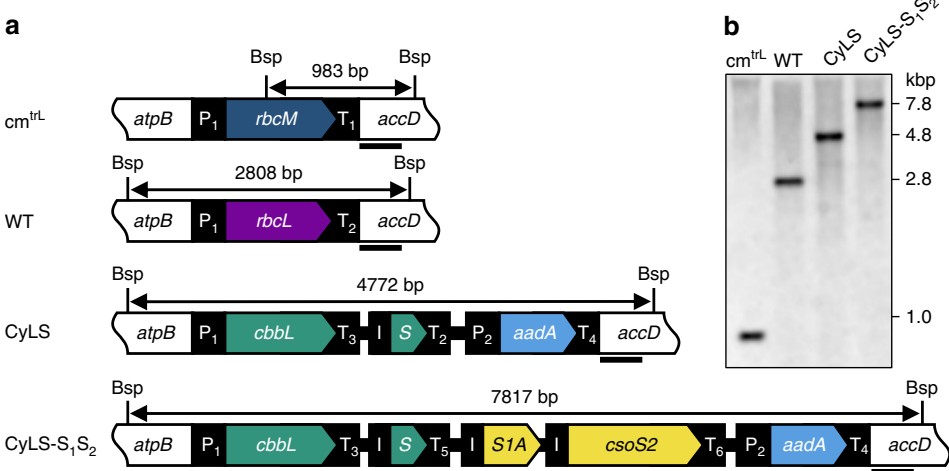

**Fig. 2** Cyanobacterial Form-1A Rubisco and carboxysome gene constructs. **a** Plastome content at the Rubisco large subunit locus in the cm^trL (*Rhodospirillum rubrum* Rubisco—*rbcM*) master line[37], wild-type tobacco (WT) and transformant plants (CyLS and CyLS-S$_1$S$_2$). The recipient line, cm^trL, was transformed via homologous recombination with constructs CyLS and CyLS-S$_1$S$_2$ (GenBank accession numbers MH051814 and MH051815) containing *Cyanobium* PCC7001 Rubisco *cbbL* and *cbbS*, together with carboxysome shell gene *csoS1A* (*S1A*) and *csoS2* sequences, codon optimized for expression in *N. tabacum* chloroplasts. The *Cyanobium cbbL* sequence was also codon matched to the tobacco *rbcL* gene where there was amino acid identity. Transformation vectors also contained the *aadA* selection marker under control of the *NtPpsbA* promoter to allow growth of transformants on spectinomycin. Each transformation construct was flanked by partial plastome *atpB* and *accD* sequence, and transformants were generated using biolistic bombardment. The locations of promoters (P), terminators (T) and intercistronic expression elements (I) are indicated. P$_1$, *NtPrbcL*; P$_2$, *NtPpsbA*; T$_1$, *NtTpsbA* originally from pcmtrLA[37]; T$_2$, *NtTrbcL*; T$_3$, *AtTpetD*; T$_4$, *NtTrps16*; T$_5$, *AtTpsbA*; T$_6$, *CrTrbcL*. A DNA probe was constructed by PCR to anneal to the *accD* flanking region in each plastome (black bar), and the corresponding size of Bsp119I (Bsp) digestion fragments are shown for each genotype (in bp). **b** DNA blots of total leaf DNA digested with restriction enzyme Bsp119I and probed with the *accD* probe, indicating successful insertion of transgenes and loss of the cm^trL genotype. DNA fragment sizes in kbp are shown

from *Cyanobium* cells grown in liquid culture. Using this approach, ca. 100 nm particles were purified from CyLS-S$_1$S$_2$ leaves by sucrose density-gradient centrifugation (Supplementary Fig. 1) and were found to be isometric with native *Cyanobium* carboxysomes, as determined by TEM and particle size analysis (Fig. 3e–g, Table 1). Notably, the particles isolated from CyLS-S$_1$S$_2$ leaves had slightly less defined vertices than their authentic *Cyanobium* counterparts (Fig. 3e–f).

**Protein content in isolated carboxysomes**. Both the carboxysomes isolated from CyLS-S$_1$S$_2$ plants and genuine *Cyanobium* carboxysomes were subjected to SDS-PAGE and immunoblots to determine protein presence and identity. The protein content of plant-derived carboxysomes was consistent with the protein complement of wild-type *Cyanobium* carboxysomes (Fig. 3h). Since wild-type *Cyanobium* carboxysomes consist of at least nine polypeptides and those of CyLS-S$_1$S$_2$ plants only four, there was relatively more of each protein in plant-derived carboxysomes as a proportion of total protein (Fig. 3h). We also noted that the CyLS-S$_1$S$_2$ carboxysomes were generally of higher purity than those isolated from *Cyanobium* (Supplementary Fig. 2). Nonetheless, both CyLS-S$_1$S$_2$ leaves and their isolated particles contained CbbL, CbbS, CsoS1A and CsoS2 in similar proportion to carboxysomes from *Cyanobium* (Fig. 3a, h). The complete suite of *Cyanobium* carboxysome proteins was absent from those in CyLS-S$_1$S$_2$ plants due to the minimized gene set utilized. Despite the presence of small quantities of tobacco Rubisco SSU leaf extracts of CyLS-S$_1$S$_2$ plants (Fig. 3a), it was absent from isolated particles, indicating no hybrid *Cyanobium*-tobacco Rubisco formation within the carboxysomes (Fig. 3h).

We identified CsoS2 as a single protein isoform in both *Cyanobium* carboxysomes and CyLS-S$_1$S$_2$ particles (Fig. 3a, h). The SDS-PAGE banding pattern of CsoS2 in CyLS-S$_1$S$_2$ plant

extracts is indicative of potential degradation in the leaf (Fig. 3a), but notably the relatively clean band in isolated carboxysomes suggests that degraded protein is not incorporated into the carboxysome (Fig. 3h).

Using current knowledge of carboxysome protein interactions and the minimal protein set used to generate the structures found in CyLS-S$_1$S$_2$ plants, we formulated a structural model of the carboxysomes formed in these plants (Fig. 3i). This model highlights the absence of specific shell components (i.e. proposed facet proteins CsoS1D and CSoS1E and vertex proteins CsoS4A and CsoS4B) and the internal carbonic anhydrase, CsoSCA, compared to the model of the *Cyanobium* carboxysome (Fig. 1c).

**Immuno-gold localization and aberrant carboxysome formation**. Immunogold labelling of CyLS-S$_1$S$_2$ chloroplast TEM sections, using an antibody specific to the carboxysomal shell protein CsoS1A, showed an association of gold particles with the carboxysome structures (Fig. 4a–c). This confirmed that the structures observed in situ contained CsoS1A. This analysis highlighted the presence of occasional rod-shaped particles in chloroplasts that also reacted positively to CsoS1A antibodies. Approximately 4% of carboxysomes observed in chloroplast sections appeared to be elongated rods (Fig. 4c–f, Table 1). These rod-shaped structures were also present in isolated carboxysome fractions (Fig. 4) at a rate of approximately 16% of purified carboxysome particles (Table 1). They were of variable length but of regular diameter [59 ± 5 (s.d.) nm], with sub-structural particles of ~12 nm in diameter, which we interpreted as Rubisco (Fig. 4f).

**Cyanobacterial Form-1A Rubisco-dependent plant growth**. Presuming that inclusion of the *Cyanobium* Rubisco in tobacco chloroplasts, either alone or within carboxysomes, should lead to a high $CO_2$ requirement for growth (Fig. 1), we characterized the

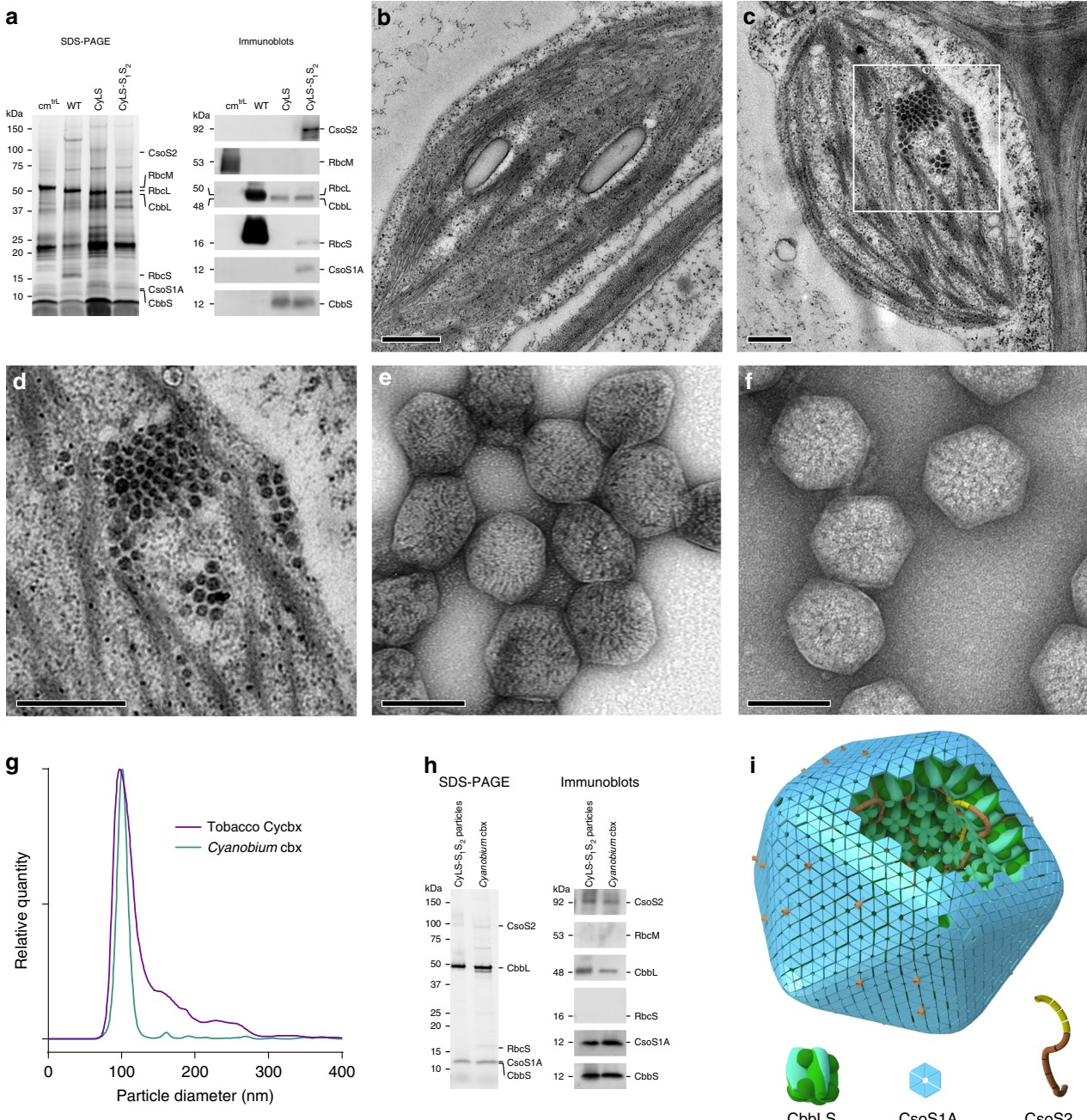

**Fig. 3** Carboxysomes are synthesized in tobacco chloroplasts from four proteins. **a** SDS-PAGE and immunoblots of Rubisco and carboxysomal proteins expressed in leaves of the recipient plant line (cm$^{trL}$), wild-type tobacco (WT) and the transformant lines CyLS and CyLS-S$_1$S$_2$. **b** Transmission electron micrographs (TEM) of chloroplasts from tobacco expressing *Cyanobium* Rubisco (CyLS plants) and tobacco expressing *Cyanobium* Rubisco along with the shell proteins CsoS1A and CsoS2 (CyLS-S$_1$S$_2$ plants) (**c**), showing aggregations of electron-dense particles of approximately 100 nm. The inset in **c** at higher magnification (**d**). Scale bars 500 nm for images **b–d**. Negatively stained carboxysomes purified from CyLS-S$_1$S$_2$ plants (**e**) and carboxysomes purified from *Cyanobium* cyanobacterial cells (**f**). Scale bars for purified carboxysomes 100 nm. **g** Diameters of carboxysomes from wild-type *Cyanobium* cells (cyan line) and carboxysomes purified from CyLS-S$_1$S$_2$ plants (magenta line) determined using a Nanosight particle analyser. **h** SDS-PAGE and immunoblots of proteins in particles isolated from CyLS-S$_1$S$_2$ leaves and genuine carboxysomes from cultured *Cyanobium* cells (*Cyanobium* cbx). **i** A model of the carboxysome structures produced in transgenic CyLS-S$_1$S$_2$ plant chloroplasts indicating the four protein components required to generate the structure. Comparison can be made with the complete wild-type structural model presented in Fig. 1c in which the pentameric vertex proteins (CsoS4AB), ancillary shell proteins (CsoS1D and CsoS1E) and carbonic anhydrase (CsoSCA) are present

Rubisco and plant CO$_2$ assimilation characteristics of transformed plants. Analysis of Rubisco catalytic performance in clarified leaf extracts of CyLS and CyLS-S$_1$S$_2$ plants revealed an enzyme with high catalytic turnover rate ($k_{cat}$) and Michaelis–Menten constant for CO$_2$ ($K_C$), consistent with values previously reported for the *Cyanobium* enzyme (Table 2 and ref. [18]). Both CyLS and CyLS-S$_1$S$_2$ plants demonstrated autotrophic growth at 2% ($v/v$) CO$_2$ (Fig. 5). Both transgenic plant types contained similar quantities of Rubisco but ten-fold less than wild-type tobacco (Table 2). For CyLS plants, leaf

**Table 1 Elongated carboxysome numbers in chloroplast cross-sections and in purified preparations**

| Particle type | Proportion in chloroplast sections[a] | Proportion in purified extracts[a] | Dimensions (nm)[b,c] |
|---|---|---|---|
| *Cyanobium* carboxysomes | n.a. | 100% | 101.3 ± 0.7 |
| CyLS-$S_1S_2$ carboxysomes | 96% ($n = 565$) | 84% ($n = 412$) | 100.3 ± 1.5 |
| CyLS-$S_1S_2$ elongated rods | 4% ($n = 24$) | 16% ($n = 78$) | 59 ± 5 ($n = 16$) × 437 ± 190 ($n = 12$) |

[a]Proportions in each particle type in chloroplast sections and purified extracts were determined from analysis of TEM images. Elongated rods were defined as structures with average widths <75 nm and disproportionate length
[b]Carboxysome dimensions were determined as the mode (±s.e.m.) of particle distributions measured using a Nanosight NS300 from 12 analyses of two independent preparations of *Cyanobium* carboxysomes and 19 analyses of three independent preparations of CyLS-$S_1S_2$ carboxysomes
[c]Elongated rod dimensions were determined as mean widths and lengths (±s.d.) of particles analysed in TEM images of purified carboxysome preparations from CyLS-$S_1S_2$ leaves. Note that there is a greater proportion of elongated particles observed in purified extracts, which we assume to be due to a combination of: breakage of very long particles during isolation, giving rise to a greater proportion of rods and, difficulty in accurately identifying elongated carboxysomes in random cross-sections of chloroplasts. n.a. not applicable

photosynthetic $CO_2$ response curves were consistent with the quantity and catalytic performance of the *Cyanobium* Rubisco, confirmed by modelling of photosynthetic rates using the Rubisco catalytic properties of the *Cyanobium* enzyme (Fig. 5, Table 2). However, the very low rates of $CO_2$ fixation by both transformants at the $CO_2$ concentrations provided in gas exchange experiments (Fig. 5a, b), coupled with the high $K_C$ of their Rubisco and its potential progressive deactivation as $CO_2$ decreases in the gas-exchange chamber, provided potentially misleading information about true photosynthetic performance of these plants. To gather more information regarding the $CO_2$ assimilation phenotypes of CyLS and CyLS-$S_1S_2$ plants, we conducted measurements at high $CO_2$ using a membrane inlet mass spectrometer (MIMS).

Photosynthetic $CO_2$ responses of leaf discs by MIMS enabled analysis of photosynthetic assimilation rates up to 20 mbar $CO_2$ (~2% $v/v$, Table 2; Fig. 5c), mimicking physiological conditions that are predicted in a fully functional chloroplastic CCM. These conditions indicated a capacity for CyLS plants to reach similar photosynthetic rates to their wild-type counterparts (Table 2; Fig. 5c) despite lower Rubisco content, consistent with the catalytic properties and content of the cyanobacterial Rubisco. MIMS analysis of CyLS-$S_1S_2$ plants revealed lower $CO_2$ assimilation rates at 20 mbar $CO_2$ (Table 2; Fig. 5c). This was consistently lower than mathematical modelling would indicate, based on Rubisco content and catalysis. CyLS-$S_1S_2$ plants also took considerably longer to reach maturity (Fig. 5d–j).

**Catalytic parameters of isolated carboxysomes.** To investigate the role that Rubisco encapsulation might play in the $CO_2$ assimilation phenotype of the CyLS-$S_1S_2$ plants, carboxysomes were isolated from plant tissue and their functionality compared with those of isolated *Cyanobium* carboxysomes and the free CyLS Rubisco. In this analysis, an approximately two-fold higher Michaelis–Menten constant for RuBP ($K_{MRuBP}$) was observed in both the plant-derived and wild-type *Cyanobium* carboxysomes compared with the free Rubisco form (Table 2, Supplementary Fig 3), consistent with resistance to access of RuBP via the carboxysome shell.

We assessed the potential contribution of the carboxysome shell and the carboxysomal CA (CsoSCA) to Rubisco performance by carrying out $CO_2$ response assays on the free CyLS Rubisco, isolated CyLS-$S_1S_2$ carboxysomes and *Cyanobium* carboxysomes, in the presence or absence of the CA inhibitor acetazolamide (AZ; Supplementary Fig. 3). This showed that there was no observable resistance to $CO_2$ influx by the carboxysome shell, with the free *Cyanobium* enzyme and the enzyme in both carboxysome types having similar $K_C$ values (Table 2, Supplementary Fig. 3). The inhibitor AZ also had no observable effect on $CO_2$ fixation rates in the *Cyanobium* carboxysomes that contain a CA enzyme and no effect on either

the free Rubisco or the CyLS-$S_1S_2$ Rubisco (Supplementary Fig. 3).

We found with measurements of both $CO_2$ and RuBP supply to the enzyme that the plant-derived carboxysomes showed a significantly lower $k_{cat}$ than either of its counterparts (Table 2, Supplementary Fig. 3), indicating that a sizable percentage of internalized Rubisco active sites were not capable of the expected rate of catalysis compared to naked Rubisco.

To determine whether the low $k_{cat}$ of the CyLS-$S_1S_2$ carboxysomal Rubisco resulted from incorrect formation of $L_8S_8$ holoenzymes, we assessed the relative stoichiometry of Rubisco isolated from CyLS plants with that found in enriched CyLS-$S_1S_2$ and wild-type *Cyanobium* carboxysomes using western blots (Supplementary Fig. 4). This confirmed that the plant-derived carboxysomes contained Rubisco with stoichiometry not significantly different to that of either the free enzyme from CyLS plants or that of *Cyanobium* carboxysomes.

Assuming that the catalytic phenotype of the plant-derived carboxysomes might result from inactivation of CyLS-$S_1S_2$ carboxysomal Rubisco, we attempted to compare the catalytic performance of Rubisco in intact carboxysomes with that of the free enzyme after freeze–thaw treatment[38] commonly used to break the structures and release free Rubisco. While this was extremely successful for *Cyanobium* carboxysomes, we could not achieve significant rupture of carboxysomes from CyLS-$S_1S_2$ plants (Supplementary Fig. 2). Instead, to examine the relative performance of free Rubisco compared to that of carboxysome-encapsulated Rubisco from CyLS-$S_1S_2$ leaf extracts, we used the supernatant fraction of crude leaf homogenates after high-speed centrifugation that contained Rubisco but was depleted in carboxysome proteins (Supplementary Fig. 5). This revealed that the free Rubisco from CyLS-$S_1S_2$ plants had both a lower $K_{MRuBP}$ and a higher $k_{cat}$ than its encapsulated counterpart (Table 2). Rubisco in broken *Cyanobium* carboxysome preparations maintained a high $k_{cat}$, but its $K_{MRuBP}$ was lower in the absence of the carboxysome shell (Table 2, Supplementary Fig. 3). As $K_C$ values did not appear to be diagnostic of carboxysome function in our assays, it was not determined for broken *Cyanobium* carboxysomes (Table 2).

**Discussion**

The expression of structurally intact carboxysomes within a $C_3$ plant chloroplast is a critical and complex engineering milestone towards the longer-term goal of attaining a functional chloroplastic CCM in $C_3$ crop plants[4,5]. The structures reported here mimic the gross structure of carboxysomes from *Cyanobium* but lack specific components expected to be required for full functionality in an operating CCM (viz. the CsoSCA, vertex proteins—CsoS4A and CsoS4B and potential metabolite-pore shell components CsoS1D and CsoS1E; cf. Figs. 1c and 3i). Despite these missing components, we show that a simplified set of proteins (consisting of the major

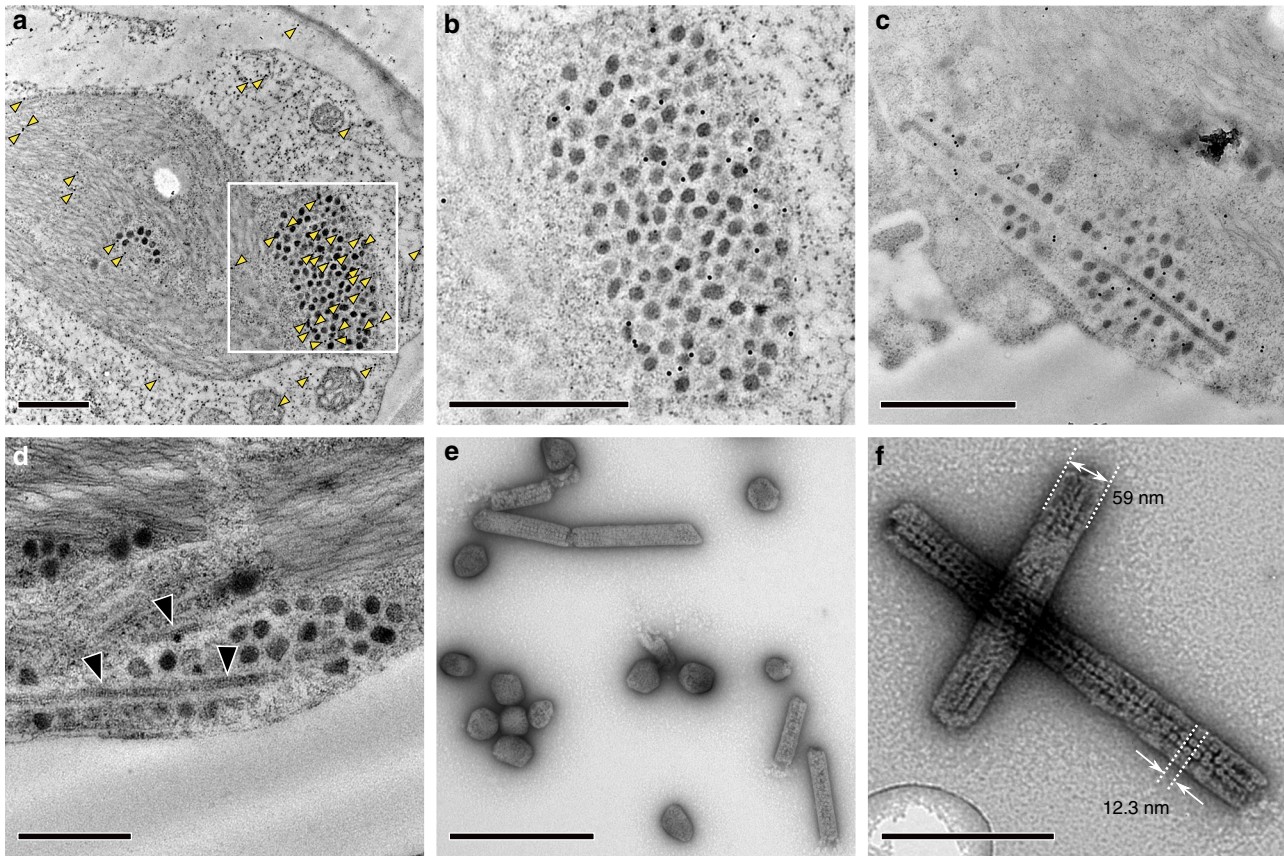

**Fig. 4** Elongated structures co-purify with plant carboxysomes. The localization of the *Cyanobium* carboxysome shell protein CsoS1A was determined in ultrathin sections using a primary antibody specific to *Cyanobium* CsoS1A and secondary antibodies conjugated with 10 nm gold particles. Gold particles are indicated by yellow arrowheads in **a** and the inset at higher magnification can be seen in greater detail (**b**). Gold particles show a co-association with both carboxysome-like structures (**a**, **b**) and elongated rods (**c**) in CyLS-S₁S₂ plants. Scale bars 1.0 μm. **d** TEM of ultrathin sections through CyLS-S₁S₂ plant chloroplasts reveal the presence of elongated structures (arrowheads) associated with the more regular ca. 100 nm carboxysome structures. These elongated structures co-purify with the plant-expressed carboxysomes (**e**) on sucrose gradients and are observed to be of variable length but of regular width of 59 ± 5 (s.d.) nm (Table 1). These structures are consistent with observed rod-shaped carboxysomes in vertex protein deletion mutants[34]. Substructure (**f**) of the rod-shaped particles shows doughnut-shaped regions of 12.3 ± 0.6 (s.d.) nm periodicity, reminiscent of Rubisco holoenzymes observed in purified *Halothiobacillus neapolitanus* carboxysome preparations[67]. Scale bars **a–e** 500 nm, **f** 200 nm

outer-shell component CsoS1A, the Rubisco linker CsoS2 and Rubisco itself) are capable of self-organizing to produce structural carboxysomes. This finding is an advance on previous attempts to generate either β-carboxysome shells[26] or interlinked Rubiscos[24] in chloroplasts and demonstrates a way forward to functional carboxysome construction in C₃ plants.

Simplified carboxysomes in CyLS-S₁S₂ plants resembled those from *Cyanobium* but with predictable differences. Occasional elongated carboxysome structures in tobacco chloroplasts are consistent with observed structures resulting from the deletion of *csoS4* vertex protein genes in *Halothiobacillus neapolitanus*[34]. A similar phenotype is found in β-carboxysomes where a genetic lesion interrupts expression of the vertex protein homologue CcmL[39]. Vertex protein mutants in both carboxysome types are functionally hindered, probably due to leakiness of the normally gas-tight shell[34,39]. We therefore conclude that the elongated structures observed in CyLS-S₁S₂ plants and extracts result from the absence of *csoS4A* and *csoS4B* vertex protein genes in the expression cassette. The observed slightly rounded shape of CyLS-S₁S₂ carboxysomes (Fig. 3e) is possibly attributable to the absence of the CsoS1D protein, which is predicted to be a large metabolite, gated pore in the mature α-carboxysome shell[30] and a likely contributor to shell rigidity[27].

The generation of carboxysomes with just four proteins represents a pivotal step in our understanding of α-carboxysome biogenesis. Despite the need for additional proteins to produce structurally identical carboxysomes to those of *Cyanobium*, as few as four proteins are required to make an encapsulating body. Thus, relatively simple rules lead to the self-assembly of icosahedral protein bodies, of the correct size, containing Rubisco. This confirms our hypothesis that CsoS1A, CsoS2 and Rubisco alone are required for simple carboxysome formation. More complex requirements for α-carboxysome structure have been assumed in the past. For example, construction of the α-carboxysomes of *H. neapolitanus* in *Escherichia coli* utilized an almost complete carboxysome operon of ten genes to achieve structural formation[27]. Our results add further weight to a streamlined approach to carboxysome construction in plants, such as the synthetic domain-fusion approach described for β-carboxysomes[40], but highlights the existence of already relatively simple gene sets in some cyanobacteria for this purpose.

The relative simplicity of the *Cyanobium* carboxysome makes it a suitable candidate for expression in chloroplasts. The utility of *Cyanobium* as a genetic donor is further emphasized by the formation of a single gene product from *csoS2* (Fig. 3a, h[41]). CsoS2 plays a critical role in carboxysome formation in

**Table 2 Rubisco catalytic properties, leaf Rubisco content and photosynthetic performance**

| Rubisco source | $k_{cat}$ (s$^{-1}$) | $K_{MRuBP}$ (μM) | $K_C^{N_2}$ (μM) | $K_C^{21\%O_2}$ (μM) | $S_{C/O}$ (M M$^{-1}$) | Rubisco (μmol sites m$^{-2}$) | Assimilation rate at 20 mbar $CO_2$ (μmol m$^{-2}$ s$^{-1}$) | $CO_2$ compensation point (Γ, μbar) |
|---|---|---|---|---|---|---|---|---|
| Tobacco | 3.1 ± 0.3 | 19 ± 3 | 9.7 ± 0.1 [290 ± 3] | 18.3 [548] | 80 ± 2.6 [2123 ± 69] (n = 4) | 21.9 ± 0.7 (n = 3) | 27.6 ± 0.5 (n = 3) | 55 ± 1 (n = 3) |
| CyLS plants | 9.8 ± 0.2 (n = 8) | 38 ± 1 (n = 9) | 158 ± 8 [4724 ± 249] (n = 4) | 275 ± 8 [8234 ± 240] (n = 3) | 55 ± 1 [1445 ± 19] (n = 3) | 2.1 ± 0.3 (n = 3) | 24.7 ± 2.3 (n = 6) | 503 ± 69 (n = 3) |
| CyLS-S$_1$S$_2$ plants | | | | | | | | |
| Free Rubisco | 9.4 ± 0.4 (n = 7) | 36 ± 1 (n = 3) | 169 ± 14 [5063 ± 416] (n = 4) | 285 ± 13 [8533 ± 389] (n = 3) | n.d. | 1.5 ± 0.3 (n = 3) | 8.1 ± 0.4 (n = 8) | 553 ± 64 (n = 3) |
| Isolated carboxysomes | 4.9 ± 0.3 (n = 3) | 59 ± 3 (n = 6) | n.d. | 248 ± 21 [7430 ± 629] (n = 3) | n.d. | n.a. | n.a. | n.a. |
| Cyanobium cells | | | | | | | | |
| Isolated carboxysomes | 9.5 ± 0.3 (n = 3) | 79 ± 2 (n = 9) | n.d. | 242 ± 21 [7250 ± 629] (n = 3) | n.d. | n.a. | n.a. | n.a. |
| Broken carboxysomes | 10.4 ± 0.1 (n = 3) | 39 ± 2 (n = 6) | n.d. | n.d. | n.d. | n.a. | n.a. | n.a. |

Leaf samples of wild-type tobacco plants, CyLS and CyLS-S$_1$S$_2$ plants, grown at 2% $CO_2$ (v/v), were extracted and prepared for Rubisco catalytic analysis as described in Methods. Tobacco $k_{cat}$ and $K_C$ values are from Sharwood et al.[70] and $K_{MRuBP}$ is from Whitney et al.[74]. Free Rubisco from CyLS-S$_1$S$_2$ plants was obtained after high-speed centrifugation of crude leaf homogenates to pellet insoluble carboxysomes. Carboxysomes from both CyLS-S$_1$S$_2$ plants and *Cyanobium* cells were isolated as described in Methods and those of *Cyanobium* were broken to release free Rubisco by freeze–thaw treatment. Rubisco specificity values for $CO_2$ ($S_{C/O}$) were determined as described in Methods and are in vitro estimates from wild-type tobacco and CyLS plant Rubiscos. Data are means ± s.e.m. from (n) replicate measurements. Rubisco active site content of CyLS-S$_1$S$_2$ leaves was consistently lower than for CyLS leaves, but this was not statistically significant (P = 0.068; two-tailed, homoscedastic Student's T test). Assimilation rates at 20 mbar $CO_2$ were determined for independent leaf discs via membrane inlet mass spectrometry analysis. Values in brackets are in μbar, and bar bar$^{-1}$ for $S_{C/O}$
n.d. not determined, n.a. not applicable

α-cyanobacterial carboxysomes[33] and is a highly disordered protein that contains recognizable repeat domains at the N-terminus and the middle (M) region of the protein and a unique C-terminal region that possibly protrudes through to the carboxysome exterior[33]. In some α-carboxysomes, CsoS2 occurs as two isoforms (CsoS2A and 2B). *Cyanobium*, however, produces only one form of CsoS2, due to the lack of an internal frame-shifting motif in the native gene sequence, which leads to C-terminal truncation during peptide synthesis in homologous sequences[41]. Additionally, *Cyanobium* carboxysomes require only one CsoS1 shell protein (Fig. 1c[29]). The model α-carboxysome from *H. neapolitanus* has three CsoS1 shell proteins (CsoS1A, B and C)[36], as does its β-carboxysome counterpart from *Synechococcus elongatus* PCC7942[42]. The α-carboxysome operons of the oceanic α-cyanobacteria are all relatively simple[29], while the *Cyanobium* Rubisco has kinetic parameters that approach those of the β-cyanobacteria[18], and outpace those found in its α-cyanobacterial *Prochlorococcus* relatives[43]. Taken together, our results point to *Cyanobium* carboxysomes as ideal practical components for use in a chloroplastic CCM.

The ability to isolate and analyse purified plant-generated carboxysomes with relative ease is important. The isolation of highly pure carboxysome fractions of α-carboxysomes[38] enables aspects of their structure and functionality to be addressed in vitro[44]. This ensures that analysis can confirm carboxysome function in C$_3$ chloroplasts in the absence of the functional $HCO_3^-$ transporters required to generate a working chloroplastic CCM[4]. This is likely to save considerable time within the chloroplastic CCM engineering strategy. Here we have demonstrated the capability to analyse and evaluate aspects of function in our plant-expressed carboxysomes in comparison with those of *Cyanobium*.

Autotrophic growth of our transformed plants demonstrates the first example of C$_3$ plants reliant on a high-catalytic-rate Form-1A Rubisco. The α-carboxysomal Form-1A Rubisco is phyletically distant from the Form-1B isoform utilized by terrestrial plants, having arisen outside the plant lineage[45]. Limited data are currently available for Form-1A Rubisco catalytic

properties, and this report highlights that further research in this area may give rise to alternative sources of Rubisco for C$_3$ photosynthesis augmentation. Furthermore, similar rates of photosynthesis in wild-type and CyLS leaf discs at 20 mbar $CO_2$ (Table 2; Fig. 5c), despite 90% less Rubisco protein in CyLS leaves, highlights the potential to achieve photosynthetic performance with appreciable reduction in nitrogen investment towards Rubisco.

While there was no significant difference in *Cyanobium* Rubisco content between the two transformant lines, we consistently found a slightly lower content in the leaves of CyLS-S$_1$S$_2$ plants (Table 2) and lower assimilation rates at 20 mbar $CO_2$ in MIMS assays (Fig. 5c). However, the assimilation rates of CyLS-S$_1$S$_2$ plants was lower than predicted by modelling (Fig. 5c), based on the measured Rubisco content in crude leaf homogenates, and maximum catalytic properties of the naked enzyme (Table 2). Importantly, the density of carboxysomes in leaf extracts leads to the separation of two populations of Rubisco in CyLS-S$_1$S$_2$ plants upon centrifugation (viz. insoluble carboxysomal Rubisco and free Rubisco). Thus observed maximum catalytic properties for CyLS-S$_1$S$_2$ Rubisco represents those of the naked enzyme, resulting from either a population of unpacked Rubisco or loss from carboxysomes during homogenization. On the other hand, our Rubisco content measurements from crude homogenates are a true representation of the total active sites. We find, however, that the $CO_2$ assimilation rate of the CyLS-S$_1$S$_2$ plants does not achieve expected rates based on both Rubisco content and maximum kinetics, leading to the conclusion that there is some kinetic impairment of the encapsulated Rubisco population. Currently, we cannot easily determine the relative quantities of Rubisco in each population due to difficulties in knowing the relative breakage of carboxysomes in extracts. In cyanobacteria, a proportion of Rubisco does appear to be in the free form, but this is difficult to measure with any degree of confidence[46].

Based on MIMS analysis, it seems unlikely that the CyLS-S$_1$S$_2$ photosynthetic phenotype results from carboxysome-mediated $CO_2$ diffusional limitation, since atmospheric $CO_2$ concentrations around 2% (v/v) are generally sufficient to overcome mutations

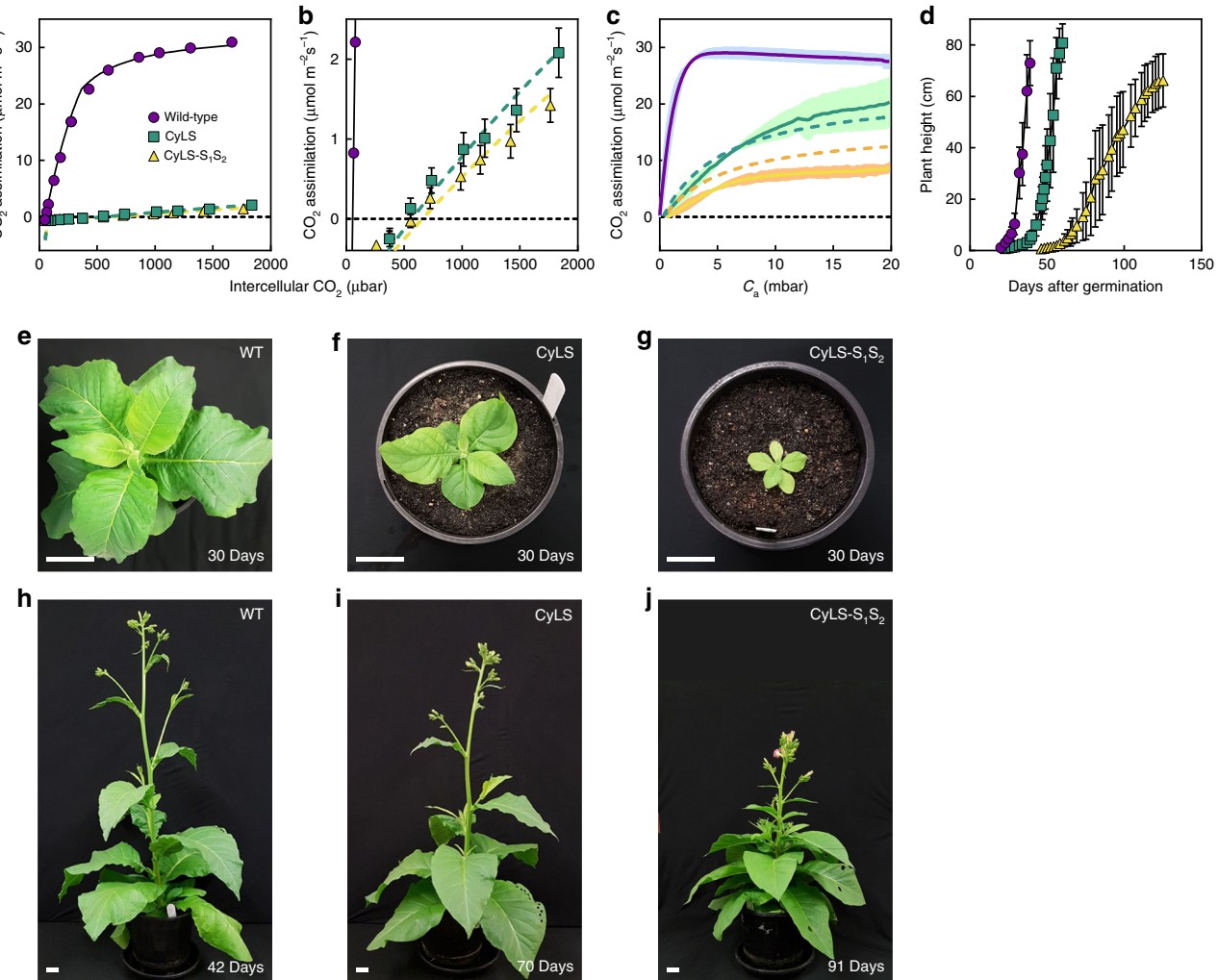

**Fig. 5** Form-1A Rubisco-dependent plant growth. $CO_2$ assimilation rates of wild-type and transgenic tobacco expressing *Cyanobium* PCC7001 Form-1A Rubisco (CyLS) and expressing *Cyanobium* Rubisco together with the carboxysome genes *csoS1A* and *csoS2* (CyLS-S$_1$S$_2$), determined by gas exchange of attached leaves. Rates are expressed on a leaf area basis (**a**) with an expanded scale for the same data presented in **b** to show assimilation rates in transformed plants. Data are presented as means ($n = $ 3-6) ± s.e.m. Fitted lines (WT, black; CyLS, cyan; CyLS-S$_1$S$_2$, yellow) were calculated as the minimum of modelled Rubisco- and electron transport-limited rates of $CO_2$ assimilation according to Farquhar et al.[64] using the Rubisco catalytic parameters presented in Table 2, and a $J_{max}$ of 140 μmol m$^{-2}$ s$^{-1}$, at 25 °C. The model predicts Rubisco active site concentrations of 1.8 ± 0.4 (s.e.m.) and 1.7 ± 0.4 (s.e.m.) μmol sites m$^{-2}$ for CyLS and CyLS-S$_1$S$_2$ leaves, respectively, within the range determined from leaf tissue (Table 2). Equations are outlined in Methods. **c** $CO_2$ assimilation rates of leaf discs from each plant line from plants grown at 2% (*v/v*) $CO_2$ in membrane inlet mass spectrometer (MIMS) assays, carried out as described in Methods. Solid lines (WT, magenta; CyLS, cyan; CyLS-S$_1$S$_2$, yellow) are averaged data from $n = 3$ independent leaf discs (±s.d., shaded areas). The dashed lines for CyLS and CyLS-S$_1$S$_2$ are modelled assimilation rates using the same parameters as **a**, **b**. Using the model to estimate $k_{cat}$ for the Rubisco in CyLS-S$_1$S$_2$ leaf discs gives an estimate of 6.46 s$^{-1}$ ± 0.01 (s.e.m.) based on the measured Rubisco content in Table 2. **d** Growth measured as plant height post germination (±s.e.m., $n = $ 4-6) for wild-type, CyLS and CyLS-S$_1$S$_2$ plants grown at 2% (*v/v*) $CO_2$. Growth phenotypes at 30 days after germination (**e**–**g**) and at maturity (**h**–**j**) of wild-type (**e**, **h**), CyLS (**f**, **i**) and CyLS-S$_1$S$_2$ (**g**, **j**) plants grown in soil at 2% (*v/v*) $CO_2$ in 20 cm pots. Note the delayed germination and time to reach maturity in both transformant lines. Scale bars 5 cm

limiting carboxysome function in cyanobacteria. For example, mutations leading to carboxysome shell protein or CA loss (but maintenance of Rubisco encapsulation) result in phenotypes that can be overcome by growth at high $CO_2$[42,47,48]. Additionally, cyanobacterial Rubisco in aberrant carboxysomes is quickly activated at high $CO_2$[49]. We also found that both CyLS and CyLS-S$_1$S$_2$ plants achieved maximum photosynthetic rates more rapidly than wild-type tobacco (Supplementary Fig. 6), indicating rapid activation of *Cyanobium* Rubisco in tobacco chloroplasts. Thus we reasoned that Rubisco activation status was both unlikely to provide an explanation for CyLS-S$_1$S$_2$ plant $CO_2$ assimilation rate

nor provide meaningful results in this case. Instead, we determined whether the catalytic properties of the Rubisco in plant-derived carboxysomes were impaired relative to the naked enzyme or that of wild-type carboxysomes isolated from *Cyanobium*. This revealed that Rubisco in plant-derived carboxysomes, lacking many of the protein factors found in wild-type carboxysomes, had a lower $k_{cat}$ than either their naked counterpart or that found in *Cyanobium* carboxysomes (Table 2). In addition, a lower $k_{cat}$ was predicted from modelling observed $CO_2$ assimilation in MIMS assays (Fig. 5). A similar observation was made by Occhialini and co-workers[50] where they successfully

co-expressed the cyanobacterial Form-1B Rubisco from *S. elongatus* with its cognate carboxysome-binding protein CcmM35 in tobacco chloroplasts. In that instance, a decrease in $k_{cat}$ was also observed, although they attributed this to an association with tobacco Rubisco SSUs[50]. We did not find any tobacco SSU associated with our purified CyLS-$S_1S_2$ carboxysomes (Fig. 3h). However, when we attempted to isolate free Rubisco from CyLS-$S_1S_2$ carboxysomes using the freeze–thaw technique[38] we were not successful (Supplemental Fig. 2). We speculate that this is due to incomplete composition of our minimal carboxysomes leading to different physical characteristics and changed Rubisco kinetics. This implies that our minimal carboxysomes lack a factor, or factors, required to ensure correct carboxysome structure and Rubisco catalytic performance. Future step-wise addition of other carboxysome proteins should enable an increase in catalytic performance to levels observed for the wild-type *Cyanobium* carboxysomes.

We also found that both CyLS-$S_1S_2$ and *Cyanobium* carboxysome Rubisco had an elevated $K_{MRuBP}$ (Table 2, Supplementary Fig. 3), indicative of resistance to substrate influx in both carboxysome types. This suggests plant-derived carboxysomes are relatively intact and that the absence of the vertex proteins does not necessarily result in substantial changes in permeability to RuBP. The observed increase in $K_{MRuBP}$ is unlikely to explain the CyLS-$S_1S_2$ plant $CO_2$ assimilation phenotype alone, although we cannot rule out the possibility that this diffusional resistance effect might be magnified in the observed clusters of carboxysomes inside chloroplasts (Figs. 3 and 4). This clustering is reminiscent of carboxysomes in β-cyanobacterial mutants lacking the ability to properly distribute their carboxysomes[42]. We hypothesize that an even distribution of carboxysomes within the stroma is preferable to prevent localized concentration gradients that could limit substrate access to carboxysomes.

We could not detect a predicted resistance to $CO_2$ flux across the carboxysome shell[12,13] as assessed by no observable change in $K_C$ for intact carboxysomes compared with the free enzyme (Table 2). We also found that the CA inhibitor AZ had no effect on carboxysomal Rubisco kinetics in our assays (Supplementary Fig. 3). We propose that this results from an inability to reproduce, in vitro, the $CO_2$:$HCO_3^-$ disequilibrium, which is normally achieved in cyanobacterial cells[11–13]. Previous modelling suggested that, under standard Rubisco assay conditions at pH 8.0, the $CO_2$:$HCO_3^-$ ratio (~0.015) would lead to negligible differences in the observed carboxylation rates by naked and carboxysomal Rubisco[51]. Thus we assume that, even at low $CO_2$ concentrations in a typical Rubisco $CO_2$ response assay, there is sufficient $CO_2$ present to overcome carboxysome shell resistance and the need for an internal CA in the carboxysome preparations described here. Further development of functional assays is needed to investigate the true properties of the carboxysome shell.

Both the marginally lower Rubisco content and poor growth phenotype of CyLS-$S_1S_2$ plants might also be indicative of impairment of protein expression. A recent study revealed the detrimental effects of overusing intercistronic expression elements (IEEs) in plastome transformation constructs[52] and here we have utilized multiple IEE sequences in the CyLS-$S_1S_2$ gene cassette (Fig. 2), based on successful gene constructs used for β-carboxysomal protein expression[24]. While this possibility does not explain the $CO_2$ assimilation phenotype of the CyLS-$S_1S_2$ plants, it may contribute to the slow germination and overall poor growth phenotype. This knowledge will inform future construct design to improve protein expression and maintain correct chloroplast function.

To generate fully functional carboxysomes within the chloroplast, further careful engineering is required. While there have been some reports of successful insertion and expression of small

multi-transgene operons in chloroplasts (e.g. refs. [53,54]), our stepwise construction of a simplified synthetic carboxysome operon minimizes the potential for errors in expression, transgene stability and carboxysome biogenesis. This also allows for a bottom–up approach to carboxysome construction, enabling analysis and evaluation of component functionality in a step-wise fashion. Addition of the minor shell proteins CsoS1D/E may be required to facilitate metabolite transport[30] and fine-tuning of structure[27], while both vertex proteins (CsoS4A and B) will be required to minimize rod-shaped carboxysome formation and ensure closure of a functional shell[34,55]. The CsoSCA must also be included to maintain the high carboxysomal $CO_2$ concentrations to support cyanobacterial Rubisco catalysis[38]. Requirement for Rubisco folding and activation chaperones (acRAF—α-carboxysome Rubisco Accumulation Factor[56]; and CbbX—a member of a Rubisco activase family whose proteobacterial functional homologue CbbQ/O[57,58] is found associated with proteobacterial α-carboxysomes[59]) is not yet known. We hypothesize that inclusion of one or all these factors should enhance the observed Rubisco catalysis to that found for *Cyanobium* carboxysomes. Beyond the construction of a functional carboxysome, a working CCM is ultimately dependent upon transport proteins, which can boost stromal $HCO_3^-$ concentrations to levels that enable carboxysome operation[5]. Recent advances in correct localization of inorganic carbon pumps to the chloroplast inner membrane[60,61] provide encouraging progress towards the achievement of this goal.

The work presented here not only sets a molecular baseline for the construction of a complete carboxysome as part of a chloroplastic CCM but also informs the assembly of α-carboxysomes and shows that simple structures can form with as few as four proteins. Coupled with results from a recent study on the principles of micro-compartment shell assembly[62], the work presented here also generates alternative avenues to tailor micro-compartment design based on simplified sets of genes. Our results draw attention to the use of α-carboxysomes (whose origins lie in already simplified gene operons) as a less complicated alternative to β-carboxysomes for the encapsulation of Rubisco in a working chloroplastic CCM. Despite the apparent complex nature of carboxysome construction, this advance indicates a feasible way forward to the engineering of a chloroplastic CCM and improved photosynthesis in $C_3$ crops.

## Methods

**Tobacco chloroplast transformation.** Chloroplasts of the *R. rubrum*-Rubisco tobacco master line were transformed with CyLS and CyLS-$S_1S_2$ constructs through biolistic bombardment according to Maliga and Tungsuchat-Huang[63] using of 2.5 mg of tungsten particles coated with freshly prepared plasmid DNA (10 μg). Each leaf was bombarded with 0.5 mg DNA-coated tungsten particles. This enabled recombination of the genes of interest in place of the *R. rubrum* Rubisco LSU (*rbcM*) gene locus in the plastid genome and selection using a spectinomycin-resistance marker gene (*aadA*) downstream of the genes of interest, under the control of the tobacco *psbA* promoter (Fig. 2). Successful explants were cultured on regeneration medium (Murashige and Skoog medium supplemented with 3% (*w/v*) sucrose, 0.5 mg mL$^{-1}$ spectinomycin, 1.0 mg L$^{-1}$ 6-benzylaminopurine [BAP], 1.0 mg L$^{-1}$ 1-naphthaleneacetic acid [NAA], 1.0 mg L$^{-1}$ thiamine-HCl, 100 mg L$^{-1}$ myo-inositol and solidified with 0.6% [*w/v*] agar) in controlled temperature cabinets (Thermoline, Wetherill Park, NSW, Australia) supplied with 2% (*v/v*) $CO_2$. Plantlets were assessed with western immunoblot and PCR to identify successful transformants. Homoplasmic transformants were obtained by subculturing the plantlets in regeneration medium. Homoplasmic plantlets were transferred to rooting medium (regeneration medium lacking BAP, NAA, thiamine and myo-inositol) and then to soil to grow to maturity for seed harvest.

**Plant growth analysis and gas exchange.** For growth and gas-exchange experiments, plants were grown from seed germinated in Green Wizard Premium Potting Mix™ (Scotts Australia, Bella Vista, NSW, Australia) in 20 cm pots supplemented with Osmocote Exact™ (ICL Australia & New Zealand, Bella Vista, NSW, Australia) at a rate of 4 g L$^{-1}$ potting mix. Plants were grown in a growth room located at the Research School of Biology Control Environment Facility,

Australian National University, Canberra, Australia, with a 23/22 °C day/night temperature regime, 12/12 h light/dark cycle, 60% relative humidity and 500 µmol photons $m^{-2} s^{-1}$, at 2.0% (v/v) $CO_2$.

**Gas exchange and modelling.** Assimilation rates (A; µmol $m^{-2} s^{-1}$) at 25 °C over the range of chloroplastic $CO_2$ partial pressure ($C_c$; µbar) were examined during gas exchange experiments using the portable flow-through LI-6400 gas-exchange system (LI-COR, Nebraska, USA). Data were modelled according to Farquhar et al.[64] and von Caemmerer[65] as the minimum of the following equations

$$A = \frac{k_{cat}^c.E(C_c - 0.5O/S_{C/O})}{C_c + K_C^{21\%O_2}} - R_d \quad (1)$$

$$A = \frac{(C_c - 0.5O/S_{C/O})J}{4(C_c + O/S_{C/O})} - R_d \quad (2)$$

using the Rubisco catalysis ($k_{cat}$, $s^{-1}$; $K_C^{21\%O_2}$, in µbar assuming a $CO_2$ solubility of 0.0334 M $bar^{-1}$), Rubisco leaf content parameters (E; µmol active sites $m^{-2}$) and in vivo $S_{C/O}$ values listed in Table 2. An ambient $O_2$ partial pressure (O) of 200 mbar, and a non-photorespiratory $CO_2$ release ($R_d$) of 1 µmol $m^{-2} s^{-1}$ were used. For Eq. 2, an electron transport rate (J) of 140 µmol $m^{-2} s^{-1}$ was used. In these simulations, it was assumed that $C_c \approx C_i$, the intercellular $CO_2$ partial pressure measured by gas exchange.

**Southern blot analysis.** Cellular DNA was isolated from 10 to 15 mg of leaf tissue using the DNeasy Plant Mini Kit (QIAGEN, Chadstone, VIC, Australia). This DNA (0.5 µg) was digested with FastDigest Bsp119I (Thermo Fisher Scientific, Scoresby, VIC, Australia) at 37 °C for 90 min. The digested DNA was separated on 0.9% (w/v) agarose by gel electrophoresis at 90 V for 2 h. The gel was prepared with 0.5× TBE and 1× SYBRSafe (Thermo Fisher Scientific). After electrophoresis, the gel was placed in denaturation buffer (0.5 M NaOH, 1.5 M NaCl) for 30 min with gentle agitation and rinsed with ultrapure water. Next it was placed in neutralization buffer (0.5 M Tris, 1.5 M NaCl, pH 7.5) for 30 min with gentle agitation and rinsed again with ultrapure water. DNA was transferred from the gel to Hybond-N+Membrane (GE Healthcare Life Sciences, Parramatta, NSW, Australia) by capillary blotting overnight as described[66]. DNA was then ultraviolet (UV) cross-linked to the membrane using the Spectrolinker XL-1000 UV Crosslinker (Spectronics Corporation, Westbury, NY, USA) in the Optimal Crosslink mode. The membrane was prehybridized for 2 h in 0.5 M NaCl with 4% (w/v) blocking reagent (GE Healthcare Life Sciences) in a hybridization oven at 55 °C. The probe was prepared using the AlkPhos Direct Labelling System (GE Healthcare Life Sciences) and a 483 bp PCR product generated from the *accD* gene in the 3'-flanking region of the plastome insertion locus (Fig. 2) (Primers *accD* reverse: 5'-AAAGGGCGGCTTCTCCTATG-3', *accD* forward: 5'-TGCAATTAAAC TCGGCCCAA-3'). The probe was hybridized to the membrane overnight at 55 °C. The following day, two 10 min primary washes were performed at 55 °C in the hybridization oven. The wash buffer contained 2 M urea, 0.1% SDS (w/v), 50 mM $NaH_2PO_4$ pH 7.0, 150 mM NaCl, 1 mM $MgCl_2$ and 0.2% (w/v) blocking reagent (GE Healthcare Life Sciences). Two 5 min secondary washes were performed at room temperature in 50 mM Tris with 2 mM $MgCl_2$. The blot was then placed for 1 h in the dark in 1 mM AttoPhos Fluorescent Substrate (Promega, Alexandria, NSW, Australia) and imaged with a ChemiDoc MP Imaging System (Bio-Rad, Gladesville, NSW, Australia) using Epi-blue illumination and a 530/28 filter.

**Carboxysome purification.** Carboxysomes were purified from culture-grown *Cyanobium* cells and CyLS-$S_1S_2$ plants essentially as described by So et al.[38] for α-carboxysomes, with some minor modifications. For the purification of carboxysomes from the cyanobacterium *Cyanobium*, cells were grown in 10 L BG-11 freshwater medium, sparged with air enriched with 2% (v/v) $CO_2$. Cells were collected by centrifugation (6000 × g, 10 min) and resuspended in 25 mL TEMB buffer (5 mM Tris-HCl [pH 8.0], 1 mM EDTA, 10 mM $MgCl_2$, 20 mM $NaHCO_3$) containing 0.55 M mannitol and 60 kU rLysozyme (Merck-Millipore, Bayswater, VIC, Australia). Cells were incubated at 37 °C in the dark for 2–16 h with gentle shaking to enable cell wall degradation and then collected by centrifugation as above. Cells were placed on ice and resuspended in 10 mL ice-cold TEMB for 15 min prior to three passages through an EmulsiFlex-B15 cell disruptor (Avestin, Ottawa, ON, Canada) at a homogenizing pressure of ~15,000 psi. IGEPAL CA-630 (Sigma-Aldrich, Castle Hill, NSW, Australia) was added to a final concentration of 1% (v/v), and broken cells were mixed by inversion on a rotating shaker at 4 °C for 1 h. Cell debris and unbroken cells were removed by centrifugation at 3000 × g, 1 min, and the supernatant centrifuged at 40,000 × g for 20 min to generate a crude carboxysome pellet. The pellet was washed again in 20 mL TEMB containing 1% (v/v) IGEPAL CA-630. The final pellet was resuspended in 1.5 mL TEMB and clarified by centrifugation at 3000 × g, 1 min prior to loading onto a 10–60% (v/v) linear sucrose gradient in TEMB. Gradients were centrifuged at 105,000 × g for 60 min and the milky-white band towards the bottom of the gradient was collected,

diluted in 35 mL TEMB and recentrifuged at 100,000 × g for 60 min. The final carboxysome pellet was resuspended in 500 µL TEMB prior to analysis for protein content, particle size analysis, TEM and Rubisco assays.

Purification of carboxysomes from CyLS-$S_1S_2$ plants was carried out in analogous manner except that 10 g (fresh weight) leaves were extracted in 100 mL TEMB (containing plant protease inhibitor cocktail; Sigma-Aldrich) using an Omni Mixer homogenizer (Kennesaw, GA, United States) on ice with three 15 s pulses. IGEPAL CA-630 was added to a final concentration of 1% (v/v) and extracts were mixed with gentle inversion for 60 min at 4 °C. Extracts were then filtered through a single layer of Miracloth (Merck-Millipore, Bayswater, VIC, Australia) prior to removal of heavy leaf debris and starch at 3000 × g for 1 min. The supernatant was subjected to centrifugation at 40,000 × g for 20 min and subsequent isolation procedures carried out on the resulting pellet in an identical manner to those used for wild-type *Cyanobium* carboxysomes.

Carboxysomes isolated from both CyLS-$S_1S_2$ leaves and *Cyanobium* were subjected to particle size analysis using a Nanosight NS300 apparatus (Malvern Instruments, Malvern, UK) essentially according to the manufacturer's instructions. Samples were diluted 1:10,000 in reverse-osmosis purified, de-ionized, filtered water and delivered to the instrument via syringe at a pump speed of 50 units. Particles were illuminated with a blue laser at 405 nm and the instrument operated at 25 °C. A series of five 1 min videos were collected for each sample and subsequently analysed. Video data from multiple, independent carboxysome extractions were analysed and data combined to determine a final particle diameter, reported as the estimated mode ± s.e.m.

For analysis of carboxysome catalytic performance, highly enriched carboxysome fractions obtained immediately prior to application onto sucrose gradients was used for both *Cyanobium* and CyLS-$S_1S_2$ plants. This ensured a high concentration of material with low probability of carboxysome shell breakage sometimes observed after removal from sucrose gradients[67] and enabled same-day Rubisco analysis.

**Protein analysis.** For protein analysis of whole-leaf extracts, leaves were extracted in extraction buffer [50 mM EPPS, 20 mM $NaHCO_3$, 10 mM $MgCl_2$, 1% (w/v) PVPP, 5 mM DTT] using a plastic pestle in a 1.5 mL microfuge tube. Protein samples were denatured by adding 4× Laemmli Sample Buffer (Bio-Rad) and heating at 95 °C for 10 min, and insoluble debris was removed by centrifuged at 20,000 × g for 5 min. A volume of supernatant equivalent to 5.6 $mm^2$ leaf area was loaded into 4–20% Stain-free polyacrylamide gels (Bio-Rad). Proteins were separated at 180 V for 35 min in denaturing buffer [1% (w/v) SDS; 25 mM Tris, 50 mM glycine, pH 8.3]. For carboxysome extracts from both CyLS-$S_1S_2$ plants and *Cyanobium* cells, approximately 1 µg of purified carboxysome protein was loaded onto gels. Stain-free gels were imaged using Gel Doc EZ (Bio-Rad). Notably, *Cyanobium* CsoS1A and CbbS proteins co-migrate on SDS-PAGE gels, although CsoS1A contains no tryptophan residues and cannot be visualized using Stain-free gels. As a result, images of Stain-free gels containing these proteins show only CbbS at 12 kDa. Conversely, CsoS1A is visualized well using Coomassie Blue stain, whereas CbbS stains poorly. These opposing characteristics could be used effectively to identify each protein in the absence of western blots. For western blot analysis, separated proteins on Stain-free gels were transferred to Immobilon-P polyvinylidene difluoride membranes (Merck) using a Trans-Blot Turbo apparatus (Bio-Rad) and blocked in TBS-T (50 mM Tris-HCl, pH 7.6; 150 mM NaCl, 0.1% [v/v] Tween-20) containing 5% (w/v) skim-milk powder. Blocked membranes were probed for 1 h in TBS-T with polyclonal antibodies raised against tobacco RbcL (1:10,000 dilution; gifted by S.M. Whitney), *Rhodospirillum rubrum* RbcM (1:5000 dilution; AS152955, Agrisera, Vännäs, Sweden), *H. neapolitanus* CsoS1A (1:5000 dilution; AS142760, Agrisera), *Cyanobium* CsoS2 (1:5,000 dilution; prepared by Gensript, NJ, USA) and *Cyanobium* CbbS (1:5000 dilution; prepared by Genscript), respectively. The probe signal was detected with alkaline phosphatase-conjugated anti-rabbit secondary antibody (1:5000 dilution; A3687, Sigma) and visualized using the Attophos Substrate Kit (Promega). The tobacco Rubisco antibody cross-reacts with the *Cyanobium* large subunit (CbbL) but not RbcM.

**Leaf Rubisco content and kinetic analysis.** Leaf disc samples (0.5 $cm^2$), taken at the site of gas exchange immediately after measurement, were used for Rubisco content and kinetic analysis. Radiolabelled [$^{14}$C] carboxyarabanitol-$P_2$ (CABP) was used to measure leaf Rubisco content as described[68,69]. In the case of CyLS-$S_1S_2$ plants, crude leaf homogenates, clarified only by a low-speed centrifugation step (3000 × g, 1 min), were used in the determination of Rubisco content to avoid excess losses of the insoluble carboxysomes. The same method used to measure leaf Rubisco content was applied to both leaf extracts containing isolated Rubisco and enriched carboxysomes used in activity assays. Radiolabelled $^{14}CO_2$ fixation assays were used to measure maximal carboxylase activities ($v_c$) and the Michaelis constant for $CO_2$ at both ambient $pO_2$ ($K_C^{21\%O2}$) and under nitrogen ($K_C^{N2}$) using 30–440 µM $^{14}CO_2$ at 25 °C, pH 8.04 as described by Sharwood et al.[70]. Samples containing Rubisco were typically activated for 5 min prior to assay. Catalytic turnover rates ($k_{cat}$; $s^{-1}$) were determined by dividing the $V_c^{max}$ by Rubisco content as determined by $^{14}$C-CABP binding as described[70]. Similar assays were performed using 10–800 µM RuBP at 20 mM $MgCl_2$, ambient $pO_2$ and 20 mM $NaHCO_3$ to determine $K_{MRuBP}$. In the case of CyLS-$S_1S_2$ plants, analysis of Rubisco

kinetics was carried out on what we deemed to be free Rubisco after centrifugation of leaf extracts at $20,000 \times g$, 1 min, or carboxysomal Rubisco after enrichment of carboxysomes. To generate broken *Cyanobium* carboxysomes with released Rubisco, enriched carboxysome samples were centrifuged ($20,000 \times g$, 10 min) and the supernatant discarded. Pellets were then frozen at $-20\,°C$ for at least 30 min and then quickly resuspended in TEMB using a pipette to achieve breakage.

The response of CyLS Rubisco, CyLS-$S_1S_2$ carboxysomes and *Cyanobium* carboxysomes to the CA inhibitor AZ was carried out under normal assay conditions except that 500 μM AZ was included in both activation and assay buffers, allowing incubation of samples with AZ for 5 min at 25 °C prior to assay. The same concentration of the more membrane permeable analogue EZ has been shown to reduce CsoSCA activity to 13% of maximum in preparations of affinity-purified, recombinant CsoSCA enzyme from *H. neapolitanus* carboxysomes, expressed in *E. coli*[38].

In vitro $S_{C/O}$ for *Cyanobium* Rubisco was determined for protein extracted from CyLS plants using $5–10\,cm^2$ leaf material. Protein was extracted in 50 mM EPPS, pH 8.0, 5 mM $MgCl_2$ containing plant protease inhibitor cocktail (Sigma-Aldrich) and initially purified using ion-exchange chromatography on 1 mL High Q cartridge columns (Bio-Rad). Rubisco was eluted using 2 mL elution buffer (50 mM EPPS, pH 8.0, 1 M NaCl) and further purified by size-exclusion chromatography on a Superdex 200 column equilibrated with specificity buffer (30 mM triethanolamine pH 8.3, 30 mM Mg acetate) using an ÄKTA Pure chromatography system (GE Healthcare). Fractions containing Rubisco were pooled and concentrated by centrifugation through 30 kDa molecular weight cutoff filters. Purified Rubisco was used to catalyse the production of radiolabelled and 3-PGA and 2-phosphoglycolate from $^3H$-RuBP under an atmosphere containing 500 ppm $CO_2$ in $O_2$, generated using Wösthoff precision gas-mixing pumps at 25 °C, pH 8.3. The ratio of radiolabelled products was determined after their separation by high-performance liquid chromatography and scintillation counting to estimate specificity[71].

**Membrane inlet mass spectrometry**. $CO_2$ response curves of $1.2\,cm^2$ leaf discs from wild-type tobacco, CyLS and CyLS-$S_1S_2$ plants were determined according to the methods described by Maxwell et al.[72] using a purpose-built cuvette attached to a Micromass membrane inlet mass spectrometer. Discs were taken from third leaf of the same plants used for attached-leaf gas exchange [grown at 2% (v/v) $CO_2$]. At least three discs were analysed for each plant line. $CO_2$ response was determined up to 20 mbar ambient $CO_2$ ($C_a$) and measured as $CO_2$ consumption by the mass spectrometer.

**Electron microscopy**. For chloroplast ultrastructure, leaf tissue ($2 \times 2$ mm) was cut and fixed in 2.5% (v/v) glutaraldehyde/2% (v/v) paraformaldehyde (ProSciTech, Thuringowa Central, QLD, Australia) overnight, washed three times in 0.1 M phosphate buffer (423 mM $NaH_2PO_4$, 577 mM $Na_2HPO_4$, pH 7.2) and then incubated in secondary fixative [1 % (w/v) $OsO_4$] for 4 h. Leaf tissue was then serially dehydrated and embedded in LR white resin[73] (ProSciTech). Approximately 70–80 nm ultrathin resin sections were cut and stained with 2% (w/v) uranyl acetate and lead citrate. For purified carboxysomes, protein samples in TEMB were mounted directly onto TEM grids and negatively stained with 2% (w/v) uranyl acetate. Both leaf sections and purified carboxysomes were observed at 100 kV using a Hitachi HM7100 TEM.

**Source data**. The uncropped images of gels and blots are provided as Supplementary Figs. 7–8.

## Data availability

All relevant data and plant materials are available from the authors upon request. Raw data corresponding to the figures and results described in this manuscript are available online [https://doi.org/10.25911/5b4edb4bada74]. Additional data reported in this paper are presented in the Supplementary Information. Nucleotide sequences are deposited in GenBank with accession numbers MH051814 and MH051815 for CyLS and CyLS-S1S2 tobacco lines, respectively.

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

## Acknowledgements

We thank L. M. Rourke and L. Wey for technical assistance; E. Martin Avila for advice on chloroplast transformation construct design, and S.M. Whitney for providing cm^trL tobacco seed and tobacco Rubisco antibody. We also thank M. Groszmann for critical comments on the manuscript. Figures 1c and 3i were drawn by Erin I. Walsh under commission by the authors. We acknowledge the facilities and the technical assistance of the Australian Microscopy & Microanalysis Research Facility at the Centre of Advanced Microscopy, The Australian National University. This research was supported by a sub-award from the University of Illinois as part of the Realizing Increased Photosynthetic Efficiency (RIPE) project (OPP1060461), funded by the Bill & Melinda Gates Foundation to M.R.B, G.D.P. and S.v.C. The Australian Research Council, Centre of Excellence grant for Translational Photosynthesis (CE140100015) supports R.E.S., B.D.R., N.D.N., S.v.C., S.B., M.R.B. and G.D.P. The Australian Academy of Science, Thomas Davies Research Fund (30321) also supports R.E.S.

## Author contributions

Conceptualization, B.M.L., G.D.P. and M.R.B.; methodology, B.M.L., W.Y.H. and R.E.S.; formal analysis, B.M.L., W.Y.H., R.E.S., B.D.R., S.B., M.R.B. and S.v.C.; investigation, B.M.L., W.Y.H., R.E.S., B.D.R., S.K., Y.-L.L., N.D.N., B.M., S.B., M.R.B. and S.v.C.; writing—original draft, B.M.L., W.Y.H. and B.D.R.; writing—-review and editing, B.M.L., W.Y.H., R.E.S., B.D.R., S.K., N.D.N., B.M., M.R.B., S.v.C. and G.D.P.; visualization, B.M.L.,

W.Y.H. and B.D.R.; supervision, G.D.P.; project administration, B.M.L.; funding acquisition, M.R.B., S.v.C. and G.D.P.

## Additional information

**Competing interests:** The authors declare no competing interests.

