## [Peer Review File · Nature Communications]

Reviewers' comments:

Reviewer #1 (Remarks to the Author):

The photosynthetic performance of C3 plants is largely limited by the poor (some may say lousy) kinetic properties of the carboxylating enzyme Rubisco and competition between CO₂ and O₂ for the reaction. Failure, so far, to significantly improve the performance of the enzyme by genetic manipulations and the impressive progress made since the discovery of the CO₂ concentrating mechanisms in phytoplankton diverted efforts towards engineering of CCM components in C3 plants and thereby raising the concentration of CO₂ around the enzyme. Two approaches are being taken to encapsulate Rubisco within prokaryotic or eukaryotic components, carboxysomes or pyrenoids, respectively. For various reasons to be discussed elsewhere I doubt the carboxysome venture can work to improve the growth and yield of C3 plants, despite the progress made here. The authors replaced the large subunit of Rubisco gene (chloroplast encoded) in tobacco with those encoding the large and small cyanobacterial genes from *Cyanobium* and also inserted CsoS1A and CsoS2 involved in carboxysomal formation, all in the tobacco chloroplast. This resulted in the appearance of carboxysomes like bodies encapsulating Rubisco. While this was achieved for the first time (to my best knowledge) this finding isn't really surprising as it was earlier shown that the relevant carboxysomal proteins interacts with Rubisco. I am surprised, however, by the very poor performance of the transgenic plants and find it difficult to accept the explanations provided here (see below). There can be no doubt that the paper is interesting and worth publishing but, in my opinion it fits much better to specific plant literature.

Comments:

1. Although they expressed only two genes involved in carboxysome biosynthesis most of the carboxysomes appear normal (in line 205 and Table 1 the authors indicate that only 4% of the carboxysomes showed abnormal rod shape in the EM analysis and 16% among the isolated ones). But the photosynthetic activity and growth of the transgenic plants, even under 2% CO₂ is very poor. In a lengthy discussion we are told that the vertex proteins, lack of carboxysomal carbonic anhydrase, lack of metabolite pore proteins CsoS1D and E (where is the evidence that in fact they operate as pore proteins?) and improper organization of Rubisco may be responsible for the poor performance. This is easily testable. With the ability to isolate the carboxysomes, as shown, what is the kinetic properties of carboxylation with respect to RuBP (dealing with the missing pore proteins) and CO₂ (lack of CA)? For the latter, raising the ambient level of CO₂ should overcome the problem but this is not the case (Fig. 5 a-c). Actually, though the enzyme functions less in the transgenic plants, the curve (Fig. 5 b) is still almost linear at close to 0.2 % CO₂ whereas the plants were grown at 2% CO₂. Thus, it is likely that the poor growth does not relate to any of the suggestions made!!!
2. In Figure 5, b, apparently something is wrong in the model used because all the data points are significantly below the predicted line. Possibly this reflects the problem mentioned in (1), above. What would it take to fit the model to the data?

3. In Figure 5 e-J you provided pictures of the WT and transgenic plants but showing those taken after different durations from germination!!!. Although the time after germination is shown, this is misleading. The reader gets the impression that the transgenic line and in particular CylS-S1S2 grew faster than in fact it did. Please provide the appearance of the plants at the same time after germination!!!. In Figure 5 d you provided data on the growth and it is not clear why growth transgenic plants was initiated so slow.

4. Non-scientific expressions were made throughout and wrong references are cited. A few examples include: in the Abstract, Line 16, I don't think the word rapid is the correct choice, and, please be modest, neither is "major" in line 25. In the Introduction, Line 42-3, I am not aware of any inorganic carbon transporter in the thylakoid membrane. Line 52, minimal is a relative term, clearly not the correct wording. In line 103 you provided the wrong reference (also in several other places, please recheck all the references, and with that being said, other groups also contributed to the research topic, hardly mentioned!!!). In the mentioned reference the authors characterized the carboxysomal carbonic anhydrase and provided no evidence for a reduced CO₂ transfer through the carboxysomal shell, as you proposed.

Reviewer #2 (Remarks to the Author):

There are considerable efforts to enhance crop production via engineering photosynthesis, one promising approach is the engineering of a cyanobacterial CO₂ concentrating mechanism (CCM) in C₃ plants. A key step to this is the engineering of a fully functional carboxysome in the chloroplast. In this study, Long et al. make a significant and novel step towards this goal. They successfully encapsulate Form 1A Rubisco from α -cyanobacteria by expressing a core shell protein, CsoS1A, and a Rubisco linker, CsoS2. Although β -cyanobacteria Rubisco has been successfully expressed in plants along with several other β -carboxysome proteins, to date Rubisco assembly within a carboxysome derived protein shell has not been achieved. Remarkably, in part due to the absence of other structural proteins, the formed plant carboxysome structures are highly reminiscent of cyanobacteria α -carboxysomes. Although most likely not functional due to missing components, the work provides a solid foundation for the future assembly of complete and fully functional carboxysomes in chloroplasts, a key milestone for the establishment of a cyanobacterial based CCM in C₃ plants.

The manuscript is well written with the data clearly presented. The ability to isolate plant carboxysome-like structures and make a direct biochemical comparison to isolated cyanobacteria derived carboxysomes provides strong evidence of correct self-assembly and supports the notion that the future complete assembly of functional carboxysomes is feasible. The detailed Rubisco kinetic, photosynthetic and plant growth analysis further supports this.

In addition to the manuscript being well written, the figures are elegantly presented and the methods provided are extensive with suitable detail.

Some minor comments:

Introduction: Well written and covers all of the relevant literature.

Line 59: Maybe change “carboxysomally”?

Figure 1 legend: Part of sentence is unclear: “is predicted to lead stromal [HCO₃⁻] approaching”

Line 180: “both CyLS-S1S2 leaves and their isolated particles contained CbbL, CbbS, CsoS1A and CsoS2 in similar stoichiometries to genuine carboxysomes from Cyanobium”. It looks like there may be more CsoS1A and CbbS in CyLS-S1S2 particles? Could this be due to cyanobacterial isolated carboxysomes having additional proteins therefore affecting the amount of material loaded? It might be worth giving an explanation and changing “similar stoichiometries”

Figure 5a: It is hard to see the CyLS and CyLS-S1S2 data. A zoom in of the 0-5 μmol/m²/s range of CO₂ assimilation would be useful

Figure 5b: It looks like the first CyLS-S1S2 data point is only partially shown.

Line 228: “Modelling of CO₂ assimilation rates in CyLS-S1S2 plants revealed lower than expected photosynthetic rates (Fig. 5a,b).” There appears to be a larger discrepancy between the CyLS experimental and modelled data than the CyLS-S1S2 experimental and modelled data, yet this sentence only mentions CyLS-S1S2. A zoom in of Figure 5a (and 5b) might help support this sentence.

Line 291: Unneeded comma after Fig 1c

Line 307: Sentence needs expanding.

Line 434: Spelling: SYBR Safe

Line 555: Unnecessary word: Source

Reviewer #3 (Remarks to the Author):

The authors have demonstrated conclusively that minimally functional carboxysomes containing Rubisco can be reconstituted in plants and support autotrophic growth at high CO₂. This is a major demonstration of a long-discussed concept. It makes the path toward testing the potential for improving C₃ crop photosynthesis with carboxysomes clear and reasonable. Prior work by other groups on shell proteins in plants made a significant advance that suggested the step presented here should work. However, the actual encapsulation of Rubisco is a major step and one that may be harder than initially thought given the fact that the prior shell assembly work was published 4 years ago.

My only real criticism is that it is unclear why the authors do not show any kinetic properties of purified carboxysomes. They speculate about the effect of the shell on kinetics and show Rubisco kinetics, but do not show similar values for the wild type and synthetic carboxysomes that they purified easily and to high purity. Minimally, an explanation for this should be provided. Given the potential importance of this work, it is unsatisfying to leave out these data while speculating extensively on the poor performance of the minimal carboxysome containing plants. Clustering of the minimal carboxysomes or activation of the encapsulated Rubisco seem equally likely as shell limitations and this would be easily answered. The lack of these data significantly reduce the impact of the paper. Shell properties a major issue and this paper has the opportunity to make two major advances, the encapsulation of Rubisco and essential data on shell function. Instead, they have opted for only one.

Reviewer #1 (Remarks to the Author):

1. Although they expressed only two genes involved in carboxysome biosynthesis most of the carboxysomes appear normal (in line 205 and Table 1 the authors indicate that only 4% of the carboxysomes showed abnormal rod shape in the EM analysis and 16% among the isolated ones). But the photosynthetic activity and growth of the transgenic plants, even under 2% CO₂ is very poor. In a lengthy discussion we are told that the vertex proteins, lack of carboxysomal carbonic anhydrase, lack of metabolite pore proteins CsoS1D and E (where is the evidence that in fact they operate as pore proteins?) and improper organization of Rubisco may be responsible for the poor performance. This is easily testable. With the ability to isolate the carboxysomes, as shown, what is the kinetic properties of carboxylation with respect to RuBP (dealing with the missing pore proteins) and CO₂ (lack of CA)? For the latter, raising the ambient level of CO₂ should overcome the problem but this is not the case

Response: We thank the reviewer for comments on this issue and have taken the intervening time since receiving their review to study the plant carboxysome performance in greater detail. We have conducted analysis of Rubisco performance in isolated carboxysomes, from both Cyanobium and plant-derived sources, in comparison with the naked enzyme, to assess the phenotype of the CyLS-S₁S₂ plants. In short, we consistently find a diminished maximum turnover rate (k_{cat}) of the isolated plant-derived carboxysomes (also seen in CcmM35-Rubisco complexes produced in tobacco chloroplasts – Occhialini et al. 2016), which we attribute to the lack of a factor, or factors, required to achieve maximum Rubisco catalysis. We thank the reviewer for their insight here which has enabled us to provide greater depth analysis of our transgenic plants. This has also led to the re-writing of components of the discussion section where an explanation for the phenotype of the carboxysome-producing plants is given.

(Fig. 5 a-c). Actually, though the enzyme functions less in the transgenic plants, the curve (Fig. 5 b) is still almost linear at close to 0.2 % CO₂ whereas the plants were grown at 2% CO₂. Thus, it is likely that the poor growth does not relate to any of the suggestions made!!!

Response: The reviewer makes a good point and we have made some changes to the manuscript to address this issue. Upon analysis we realize that both transformant plant types perform close to modelled predictions in gas exchange, whereas at high CO₂ in MIMS assays the true phenotype is borne out. Considering the very low rates of CO₂ assimilation in our gas exchange experiments, the potential for the Cyanobium Rubisco to progressively deactivate under these conditions as CO₂ is decreased, we suggest that our confidence in these data providing a full picture of the plant photosynthetic phenotype is relatively low in the 0-1000 μ bar range. Thus, we apply more weight to the results in MIMS assays where we can measure rates of CO₂ assimilation at CO₂ concentrations more physiologically relevant and under conditions where rates provide more meaningful measurements. This has allowed us to make more solid conclusions regarding a marginally lower than expected k_{cat} of the Cyanobium Rubisco in CyLS-S₁S₂ plants as the reason for this observed phenotype. The relevant changes to the manuscript are presented in lines 231-237 within the results section.

2. In Figure 5, b, apparently something is wrong in the model used because all the data points are significantly below the predicted line. Possibly this reflects the problem mentioned in (1), above. What would it take to fit the model to the data?

Response: We thank the reviewer for pointing out this error. We had determined gas exchange measurements from a relatively large dataset but plotted only specific data points from a recent set of experiments, while calculating the curve from an average of all data. Essentially, the modelled line was fitted to the wrong dataset for a larger number of plants with more variable Rubisco content. We have ensured now that the correct curves have been calculated from the appropriate dataset and have used the model to predict Rubisco quantities, which closely match our measured values. The change can now be seen in Fig's 5 a and b.

3. In Figure 5 e-J you provided pictures of the WT and transgenic plants but showing those taken after different durations from germination!!!. Although the time after germination is shown, this is misleading. The reader gets the impression that the transgenic line and in particular CyLS-S1S2 grew faster than in fact it did. Please provide the appearance of the plants at the same time after germination!!!. In Figure 5 d you provided data on the growth and it is not clear why growth transgenic plants was initiated so slow.

Response: We have captured new images of all plants at 30 days after germination (see Fig 5) and added the following text to the Figure caption to highlight the slow germination and growth of the transformants:

“Note the delayed germination and time to reach maturity in both transformant lines.”

Response: We have also added a phrase to line 451 where we explain the potential effects of IEEs on transformant phenotypes:

*“While this possibility does not explain the CO₂ assimilation phenotype of the CyLS-S₁S₂ plants, it may contribute to the **slow germination and overall poor growth phenotype.**”*

4. Non-scientific expressions were made throughout and wrong references are cited. A few examples include: in the Abstract, Line 16, I don't think the word rapid is the correct choice, and, please be modest, neither is “major” in line 25. In the Introduction, Line 42-3, I am not aware of any inorganic carbon transporter in the thylakoid membrane. Line 52, minimal is a relative term, clearly not the correct wording. In line 103 you provided the wrong reference (also in several other places, please recheck all the references, and with that being said, other groups also contributed to the research topic, hardly mentioned!!!). In the mentioned reference the authors characterized the carboxysomal carbonic anhydrase and provided no evidence for a reduced CO₂ transfer through the carboxysomal shell, as you proposed.

Response: We thank the reviewer for their comments and have made slight changes to the abstract and main manuscript to comply with their suggestions.

In the abstract we have included the term 'relatively rapid' (line 16) to emphasise that cyanobacterial Rubiscos have a higher catalytic turnover number compared with their C₃ counterparts.

We have changed the term 'major' to 'important' (line 25).

We would specifically like to thank the reviewer for highlighting our error in describing the CO₂ conversion mechanism on the thylakoid as an inorganic carbon transporter. This has been corrected to "... CO₂-converting complexes⁷⁻⁹ on the thylakoid membranes" (line 44) and references added (Fridlyand et al. 1996, Maeda et al. 2002 and Price 2011).

We have corrected the term 'minimal' to read 'reduced' on line 54 and added a relevant reference (Coleman et al. 1982).

The reviewer is correct that the wrong reference was used in line 103 of the original manuscript. We have now inserted references to Klein et al. 2009, Reinhold et al 1991 and Dou et al. 2008. which now appear in line 105.

Reviewer #2 (Remarks to the Author):

Minor comments:

Line 59: Maybe change "carboxysomally"?

Response: We have changed the text to read:

Line 61: "Notably, a carboxysome-encapsulated cyanobacterial Rubisco"

Figure 1 legend: Part of sentence is unclear: "is predicted to lead stromal [HCO₃⁻] approaching"

We have changed the text in the Figure caption to now read:

"However, in combination (iii), generation of high stromal HCO₃⁻ pool in the presence of functional carboxysomes, with stromal CA eliminated, is predicted to generate a stromal HCO₃⁻ concentration approaching 5 mM¹⁶ and to increase in CO₂ fixation and yield of up to 60%¹⁵."

Line 180: "both CyLS-S1S2 leaves and their isolated particles contained CbbL, CbbS, CsoS1A and CsoS2 in similar stoichiometries to genuine carboxysomes from Cyanobium". It looks like there may be more CsoS1A and CbbS in CyLS-S1S2 particles? Could this be due to cyanobacterial isolated carboxysomes having additional proteins therefore affecting the amount of material loaded? It might be worth giving an explanation and changing "similar stoichiometries"

Response: The reviewer's assessment is correct, and we have edited the relevant text to now read:

Lines 181-187:

"Noting that the wild-type Cyanobium carboxysomes consist of at least nine polypeptides and those of CyLS-S₁S₂ plants only four, there is relatively more of each protein in the plant-derived carboxysomes as a proportion of total protein (Fig. 3h). We also noted that the CyLS-S₁S₂

carboxysomes were generally of higher purity than those isolated from Cyanobium (Supplementary Fig. 2). Nonetheless, both CyLS-S₁S₂ leaves and their isolated particles contained CbbL, CbbS, CsoS1A and CsoS2 in similar proportion to genuine carboxysomes from Cyanobium (Fig. 3a,h)."

Figure 5a: It is hard to see the CyLS and CyLS-S1S2 data. A zoom in of the 0-5 $\mu\text{mol}/\text{m}^2/\text{s}$ range of CO₂ assimilation would be useful

Response: We thank the reviewer for highlighting this issue. This graph and 5b were originally presented such that 5b provides a way of expanding the data in 5a by presenting the data on a Rubisco active site basis. Nonetheless, responses during the review process essentially show this is not clear. Instead, we have taken the reviewer's suggestion and now provide Fig 5b as a contracted view of the data in Fig 5a.

Figure 5b: It looks like the first CyLS-S1S2 data point is only partially shown.

We have also expanded the Y-axis to show negative values and therefore all data for each plant line.

Line 228: "Modelling of CO₂ assimilation rates in CyLS-S1S2 plants revealed lower than expected photosynthetic rates (Fig. 5a,b)." There appears to be a larger discrepancy between the CyLS experimental and modelled data than the CyLS-S1S2 experimental and modelled data, yet this sentence only mentions CyLS-S1S2. A zoom in of Figure 5a (and 5b) might help support this sentence.

Response: We thank the reviewer for highlighting this problem. It was also pointed out by other reviewers. We had plotted the model line based on additional plant measurements which were not representative of the Rubisco content of the plants actually used to plot the data points. We have now corrected this error by ensuring the model is based on the data presented. Along with axis scale changes the graph is now much clearer.

Line 291: Unneeded comma after Fig 1c

Response: Comma removed

Line 307: Sentence needs expanding.

Response: The sentence now reads:

Lines 356-358: "The photosynthetic performance of our transgenic plants was as we predicted with CO₂ response curves primarily consistent with the content and catalytic properties of the Cyanobium Rubisco (Table 2, Fig. 5)."

Line 434: Spelling: SYBR Safe

Response: Corrected

Line 555: Unnecessary word: Source

Response: Removed

Reviewer #3 (Remarks to the Author):

My only real criticism is that it is unclear why the authors do not show any kinetic properties of purified carboxysomes. They speculate about the effect of the shell on kinetics and show Rubisco kinetics, but do not show similar values for the wild type and synthetic carboxysomes that they purified easily and to high purity. Minimally, an explanation for this should be provided. Given the potential importance of this work, it is unsatisfying to leave out these data while speculating extensively on the poor performance of the minimal carboxysome containing plants. Clustering of the minimal carboxysomes or activation of the encapsulated Rubisco seem equally likely as shell limitations and this would be easily answered. The lack of these data significantly reduce the impact of the paper. Shell properties a major issue and this paper has the opportunity to make two major advances, the encapsulation of Rubisco and essential data on shell function. Instead, they have opted for only one.

*Response: We thank the reviewer for their suggestion and have spent the intervening months carrying out the experiments they have highlighted, including catalytic analysis of both wild-type *Cyanobium* carboxysomes and their plant-derived counterparts. This has led to the addition of significant detail regarding Rubisco kinetics and provides evidence for the CO₂-fixation phenotype of the carboxysome plants.*

We hope that these additional data overcome the major issue highlighted by the reviewer.

Reviewers' Comments:

Reviewer #1 (Remarks to the Author):

Reading the revised MS I am pleased to say that the reviewers successfully accomplished their task to help the authors improve the paper. The additional data provided, the modesty of the data presentation and that of the discussion and the realizations of the limitations of the approach all contributed to a far better and balanced MS. Indeed, this version is far better than the original and I am happy to recommend its publication.

Reviewer #2 (Remarks to the Author):

The authors have approached the reviewer comments in a very constructive manner. They have gone on to perform extensive further experiments on Rubisco kinetics between the different experimental lines. This data has provided some interesting insight into WT and heterologous carboxysome function, such as RuBP diffusion but not CO₂ diffusion restriction of carboxysomes. In addition, indicating that heterologous carboxysomes are potentially missing key factor(s) for fully functional Rubisco activity. I think this data further strengthens the manuscript and addresses the major concerns of the reviewers.

The authors have also satisfactorily addressed all of my minor concerns.

Reviewer #3 (Remarks to the Author):

The addition of the purified carboxysome kinetics addressed my prior concerns very well. There is still some mystery as to why they are impaired, but the possibilities have been greatly narrowed and this is very interesting. I think the interpretations of the results are very reasonable. My only comment for improvement is a suggestion that around line 83, the authors also cite Badger et al. 2002 *Funct. Plant Biol.*, 2002, 29, 161–173, where the terms alpha and beta carboxysomes were coined.

Reviewers' Comments:

We note that only Reviewer #3 had a single request:

Reviewer #3 (Remarks to the Author):

My only comment for improvement is a suggestion that around line 83, the authors also cite Badger et al. 2002 *Funct. Plant Biol.*, 2002, 29, 161–173, where the terms alpha and beta carboxysomes were coined.

We have now added this reference to the text and the reference list.